# The expanding movement of primary care physicians operating at the first line of healthcare delivery systems in sub-Saharan Africa: A scoping review

**Kéfilath Bello**[1,2]*, **Jan De Lepeleire**[3], **Jeff Kabinda M.**[4], **Samuel Bosongo**[4‡], **Jean-Paul Dossou**[1‡], **Evelyn Waweru**[2‡], **Ludwig Apers**[5‡], **Marcel Zannou**[1,6‡], **Bart Criel**[2]

1 Centre de Recherche en Reproduction Humaine et en Démographie, Cotonou, Benin, 2 Department of Public Health, Institute of Tropical Medicine, Antwerp, Belgium, 3 Department of Public Health and Primary Care, General Practice, KU Leuven—University of Leuven, Leuven, Belgium, 4 Centre de Connaissances en Santé en République Démocratique du Congo, Kinshasa, Democratic Republic of Congo, 5 Department Biomedical Sciences, Institute of Tropical Medicine, Antwerp, Belgium, 6 Faculty of Health Sciences, University of Abomey-Calavi, Abomey-Calavi, Benin

☯ These authors contributed equally to this work.
‡ These authors also contributed equally to this work.
* kefilath@gmail.com

**Data Availability Statement:** All relevant data are within the paper and its Supporting Information files.

## Abstract

### Introduction

In sub-Saharan Africa (SSA), the physicians' ratio is increasing. There are clear indications that many of them have opted to work at the first-line of healthcare delivery systems, i.e. providing primary care. This constitutes an important change in African healthcare systems where the first line has been under the responsibility of nurse-practitioners for decades. Previous reviews on primary care physicians (PCPs) in SSA focused on the specific case of family physicians in English-speaking countries. This scoping review provides a broader mapping of the PCPs' practices in SSA, beyond family physicians and including francophone Africa. For this study, we defined PCPs as medical doctors who work at the first-line of healthcare delivery and provide generalist healthcare.

### Methods

We searched five databases and identified additional sources through purposively selected websites, expert recommendations, and citation tracking. Two reviewers independently selected studies and extracted and coded the data. The findings were presented to a range of stakeholders.

### Findings

We included 81 papers, mostly related to the Republic of South Africa. Three categories of PCPs are proposed: family physicians, *"médecins généralistes communautaires"*, and general practitioners. We analysed the functioning of each along four dimensions that emerged from the data analysis: professional identity, governance, roles and activities, and output/

**Funding:** This study was done as part of the doctoral program of KB. This program is funded by a grant from the Directorate-general Development Cooperation and Humanitarian Aid (DGD), Belgium (https://diplomatie.belgium.be/en/policy/development_cooperation/who_we_are/our_organisation/dgd). Grant number: FRM 1178 v1.0 between the Institute of Tropical Medicine, Antwerp (https://www.itg.be/) and KB. The funder had no role in study design, data collection and analysis, decision to publish, or preparation of the manuscript.

**Competing interests:** The authors have declared that no competing interests exist.

**Abbreviations:** FP, family physician; GP, general practitioner; MGC, *Médecin Généraliste Communautaire*; PCP, primary care physician; PRIMAFAMED, primary care and family medicine education network; PHC, primary health care; RSA, Republic of South Africa, SSA, sub-Saharan Africa; WONCA, World Organization of Family Doctors.

outcome. Our analysis highlighted several challenges about the PCPs' governance that could threaten their effective contribution to primary care. More research is needed to investigate better the precise nature and performance of the PCPs' activities. Evidence is particularly needed for PCPs classified in the category of GPs and, more generally, PCPs in African countries other than the Republic of South Africa.

## Conclusions

This review sheds more light on the institutional, organisational and operational realities of PCPs in SSA. It also highlighted persisting gaps that remain in our understanding of the functioning and the potential of African PCPs.

## Introduction

Effective implementation of primary health care (PHC) is necessary for achieving the Sustainable Development Goals [1, 2]. In Sub-Saharan Africa (SSA), many countries operationalise PHC within health districts, which encompass a network of formal health facilities, community-based services, and other supporting services and health programs [3]. The formal healthcare delivery platform includes small to medium size public and private facilities (the first line) which should normally be the first entry point in this platform and should deliver primary care, dealing with the majority of the population's health needs. These first-line facilities (called dispensaries, health centres, community health centres or clinics, depending on the context) are supported by a district hospital (the second line) which is the first referral level. In big urban areas, the second line can also include other hospitals (usually private) or large private clinics.

Effective PHC systems need a workforce that is adequately trained, motivated, and distributed, with appropriate allocations of tasks and responsibilities [1, 4]. In most of sub-Saharan Africa (SSA), primary care, the service delivery component of PHC [5], traditionally relies on non-physician clinicians, especially nurse-practitioners [5, 6]. This resulted from national task-shifting strategies aiming to compensate for physicians' shortage and enhance the coverage and accessibility of primary care [7]. Under certain conditions, task-shifting is cost-effective for managing specific diseases and improving access to quality healthcare [7–9]. However, despite physicians' overall shortage, their ratio in SSA increased from 1.2/10000 inhabitants in 2000 to 2.3/10000 inhabitants in 2017 [10]. This increasing trend is observed in several countries [10] due to training efforts [7, 11]. For example, in Benin, physicians' ratio increased from 0.6/10000 inhabitants in 2010 [12] to 1.5/10000 inhabitants in 2017 [13]. This is likely to increase further, as roughly 90 doctors graduate each year from the country's two medical schools [14]. In the Democratic Republic of Congo, the number of doctors enrolled in the National Medical Council increased from 6000 in 2004 to 25000 in 2017, with a ratio of 1.06/10000 inhabitants in 2017 [15, 16]. Furthermore, there are disparities within countries with relatively overstaffed urban areas, where doctors seek job opportunities and compete in a limited healthcare market [14, 17].

Although the existing statistics on physicians and primary care workforce in SSA is insufficient [6], there are indications that many doctors settle at the first line [6, 17, 18], because of the government's low capacity to enrol them [6], the limited access to specialist training and, sometimes, deliberate efforts to provide medical care at this level [6, 19, 20]. In high-income countries, physicians' presence at the primary care level is common, but it is a much more

recent phenomenon in most African health systems [21, 22]. This raises several questions. How do these medical doctors function, and how are they integrated into health systems with solidly established task-shifting policies? What is their contribution to addressing the numerous challenges faced by African PHC systems, such as quality issues [14, 23], epidemiological transitions [24], and increasing expectations from the population [5, 25]? It is important to answer these questions to ensure that the presence of physicians at the first-line is not a source of disruption for the PHC systems and, to ensure the efficient use of a human resource that is still relatively insufficient.

Very few papers have synthesised the available knowledge on the physicians working at the first line of health care delivery in SSA, referred to as "primary care physicians (PCPs)" in this paper. The existing reviews were limited to the development of family medicine and the practices of family physicians [21, 22]. A scoping review published in 2020 looked at the existing evidence on family physicians' impact in SSA [22]. However, in SSA, PCPs do not necessarily have a family medicine specialisation [6, 22]. Besides, these reviews focused on English-speaking countries, even though interesting initiatives also exist in other African countries such as francophone Africa [26]. The present paper provides a broader view of the PCPs' practices at the first line of healthcare delivery in SSA. We also analysed these practices from a health system perspective.

The review seeks to address the question, "What are the main characteristics and key issues of the PCPs' practices at the first-line of healthcare delivery in SSA?". As we aimed to map the key issues related to these practices, it was most appropriate to follow a scoping review methodology.

In our review question, PCPs are defined as medical doctors who work at the first line of healthcare delivery in SSA and provide all-around care to the population, without distinction based on the age, the sex or the clinical condition. We excluded doctors who work exclusively at the hospital level because the novelty of the phenomenon we are studying lies on the shift of medical practice from hospitals to the first-line. We also limited the study to physicians who provide all-round care to the population because the key function of the first-line is to provide primary care, which is defined by the Institute of Medicine as "the provision of integrated, accessible health care services by clinicians who are accountable for addressing a large majority of personal health care needs, developing a sustained partnership with patients, and practising in the context of family and community" [27].

This paper started by proposing a categorisation of the PCPs working at the first-line and then analysed their practices using a framework that emerged from the data analysis. Finally, we discussed the implications for improving primary care in SSA.

## Methods

This scoping review followed the methodology proposed by Arksey and O'Malley [28]. Other methodological papers also guided us, including those of Tricco and al and the "Alliance for Health Policy and Systems Research" [29, 30]. A review protocol was developed and discussed among all of the authors (S1 Text).

### Identification of relevant studies

We searched the following databases: MEDLINE, Cochrane Library, Banque de Données en Santé Publique, Web of Science, and Health System Evidence. The keywords were medical doctors, first-line, primary care, and sub-Saharan Africa. Where relevant, we used MeSH and thesaurus terms. After the first round of selection in December 2018, we ended up with far too many documents for the full-text review. Therefore, we iteratively refined the search strategy.

As the majority of studies were published after 2000, we decided to narrow the publication period to 2000–2019. Another reason for starting the search in 2000 was the fact that some of the papers written after 2000 provided information on the development of the PCPs in SSA going back to earlier years. The final search strategy for MEDLINE applied on April 6, 2019, is presented in the supporting information (S1 Table). We also looked for grey literature on the websites of the World Organization of Family Doctors (WONCA) and the NGO Santé Sud. We also obtained additional sources from experts in the field of PHC and the reference lists of relevant articles.

On June 19, 2020, we performed an additional search in MEDLINE to update the studies included. Furthermore, based on the feedback from experts on this review, we performed on January 15, 2021, a search on MEDLINE by including the keywords "community doctors" and "rural doctors" in the search strategy. We finally performed a manual review of the following journals in January 2021: "Rural and Remote Health", "African Journal of Primary Care and Family Medicine", and "Human Resources for Health".

## Study selection

After removing the duplicates, two authors (KB and JK) independently screened the titles and abstracts in line with predefined selection criteria (S1 Text). The results were discussed, and the discrepancies were solved with the help of a third reviewer (SB or BC). The selection criteria were then discussed among all authors and refined as we were more familiar with the literature. The final set of selection criteria (Table 1) was then applied by the two reviewers (KB and JK) for assessing the eligibility of the full-texts. The discrepancies were again discussed and solved with the help of a third reviewer (SB or BC). For the subsequent searches, KB, SB and BC followed the same selection procedures.

## Data extraction and charting

PCPs are part of a health system's human resources, and their practices should contribute to improving the whole health system's performance. It is also likely that these practices are influenced by the other elements of the health system (governance, resources, service delivery, relationships with the community, context, values on which the system is based, and outcomes).

Table 1. Selection criteria.

|  | Inclusion criteria | Exclusion criteria |
|---|---|---|
| Type of document and period | • Peer-reviewed articles (original research, reviews)<br>• Report of interventions or research | • Opinion papers and commentaries<br>• Conference/workshop reports<br>• Papers published before 2000 |
| Content of the paper | • Papers reporting the practices of MDs in first-line health services or the community | • Papers related to MDs working exclusively in hospitals<br>• Papers related to MDs exclusively taking care of specific diseases and populations (HIV, diabetes, etc.)<br>• Papers exclusively addressing the practices of non-physicians<br>• Papers related to PHC workers in general without differentiating MDs from other cadres<br>• Guidelines<br>• Papers not related to sub-Saharan Africa<br>• Studies only assessing the knowledge of PCPs without assessing their actual practices.<br>• Studies reporting a specific experience in a controlled environment |

Therefore, we used these elements, as described in the health systems dynamics framework [31], as a starting point for constructing the data extraction form. This allowed us to systematically look at the information on each of these health system elements and analyse how they relate to the practices of PCPs.

Two reviewers (KB and SB) extracted the data from the selected studies, using the data extraction form stored in an Excel file (S2 Table). Results were compared and merged when necessary. This was the first step of data coding.

## Collating, summarising, and reporting the results

Two reviewers (KB and SB) independently revisited the extracted data and refined the coding. The resulting themes were discussed among all co-authors. Some elements of the health system dynamics framework were put together based on the strong relationships we found between them in the papers reviewed. For instance, the resources allocation and the governance arrangements were put together under the broader theme "governance" because the latter implies not only subjects like policy guidance or regulation but also the resources allocation. Other elements were renamed to reflect the reality of the PCPs' practice. The "service delivery" was thus renamed "role and activities". Furthermore, the reviewers identified additional themes emerging from the literature. One of them was the professional identity, as we realised that this theme largely determines the PCPs' practices. This new theme of professional identity regroups some themes of the health system dynamics framework such as the values underlying the practices of the PCPs, their institutional status (public or private) and the type of postgraduate training they received. It also includes another new theme: the historical pathway of the PCPs (see the results section for details).

We followed the PRISMA extension for scoping review guidelines [32] for reporting the findings (S2 Text).

## Stakeholders' consultation

We presented the preliminary findings of this review to students in the Master in Public Health at the Institute of Tropical Medicine of Antwerp in September 2019. Many of them were Africans and primary health care professionals, including primary care physicians. Also, we had informal discussions with other PhD students and African primary health care actors (from Uganda, Nigeria, Guinea, Mali for example), to get some insights on the practices of primary care physicians in their countries and refine our analysis.

# Results

## Characteristics of the included papers

The first search in electronic databases and grey literature yielded 3999 papers in total, including 35 duplicates. After screening the 3964 unique papers, based on the titles and abstracts, 3844 were excluded because they clearly did not meet the inclusion criteria. Of the remaining 120 papers, 50 met the inclusion criteria after the full-text eligibility assessment. The citation tracking and subsequent searches led to the identification of 31 additional papers meeting the inclusion criteria (Fig 1).

The 81 papers included in this review are presented in the supporting information (S3 Table). Four papers related to SSA in general, and the remaining covered 21 different countries (67% Anglophone and 33% Francophone). Southern Africa was the most represented region because most of the papers (38 out of 81) related to the Republic of South Africa (RSA). A wide range of methodologies was used in the papers and 81% were written in English (Table 2; S3 Table).

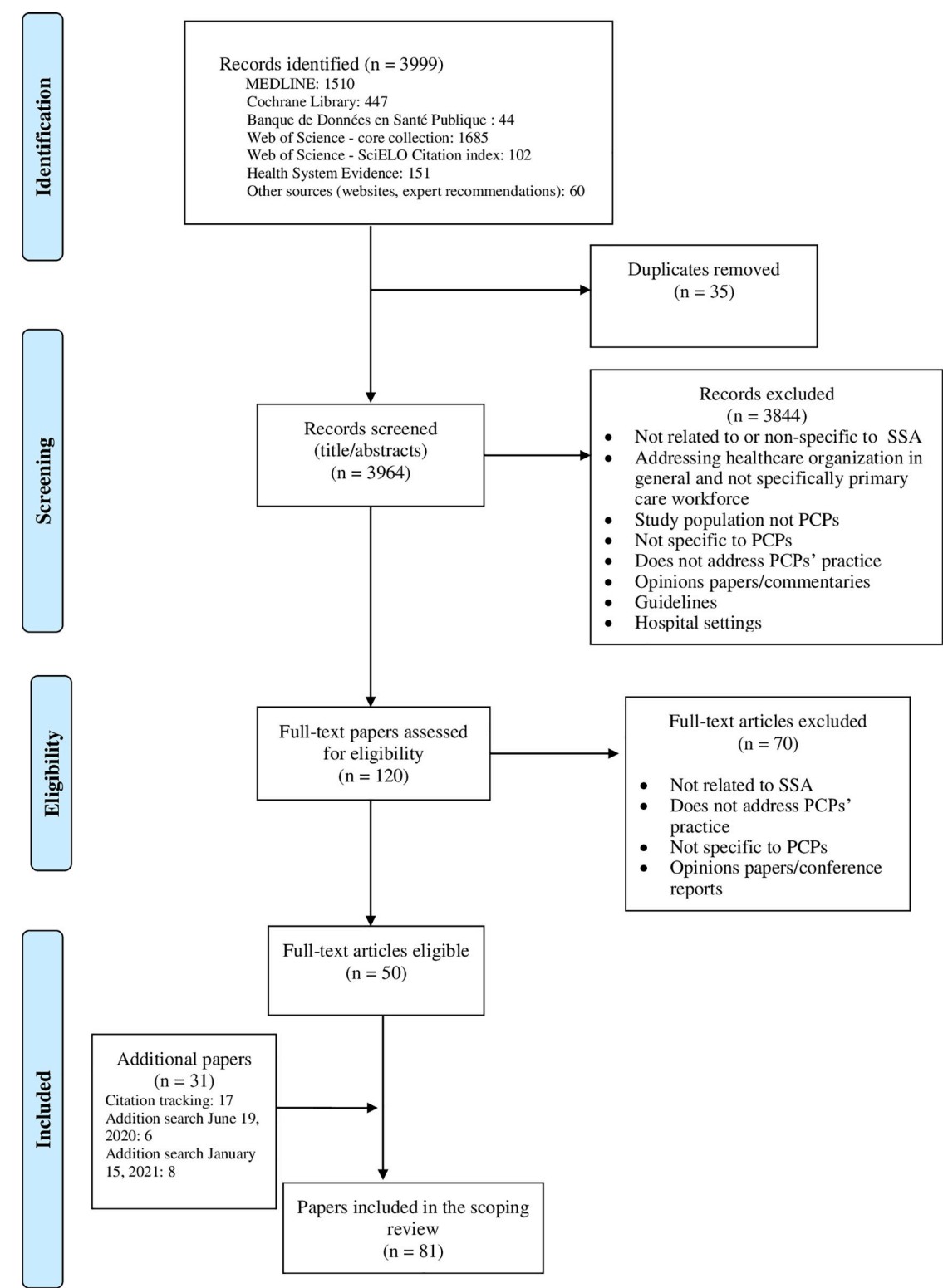

**Fig 1. Flowchart of the study selection.**

**Table 2. Characterisation of the papers included.**

| Characterisation of papers included | | Number | Percentage out of the 81 papers included |
|---|---|---|---|
| Language of the papers | English | 66 | 81% |
| | French | 15 | 19% |
| Methodology | Cross-sectional study | 24 | 30% |
| | Qualitative study | 17 | 21% |
| | Case study | 15 | 19% |
| | Mixed- methods study | 7 | 9% |
| | Narrative review | 7 | 9% |
| | Retrospective longitudinal study | 3 | 4% |
| | Action-research | 2 | 2% |
| | Delphi study | 2 | 2% |
| | Before and after study | 1 | 1% |
| | Ecological study | 1 | 1% |
| | Modelisation and stakeholders consultations | 1 | 1% |
| | Scoping review | 1 | 1% |
| Region of sub-Saharan Africa covered | Southern Africa | 50 | 62% |
| | West Africa | 26 | 32% |
| | Eastern Africa | 22 | 27% |
| | SSA in general | 4 | 5% |
| | Central Africa | 2 | 2% |

NB: Some papers related to several countries.

## From an heterogeneous nomenclature to three distinct categories

This review indicated great variety in the nomenclature used to refer to PCPs working at the first-line in SSA (Table 3).

A closer look at these names led us to an initial regrouping based on semantic similarities, which led to seven groups. We then conducted a second regrouping based on the realities described in the selected papers, which led to three major categories (Fig 2).

**Table 3. Nomenclatures of primary care physicians in sub-Saharan Africa.**

| English nomenclature | Number of papers | French nomenclature | Number of papers |
|---|---|---|---|
| Family physician | 35 | Médecin généraliste | 6 |
| General practitioner | 12 | Médecin de campagne | 5 |
| Medical Officer | 7 | Médecin généraliste communautaire | 3 |
| Primary care physician | 4 | Médecin | 2 |
| Doctor | 3 | Médecin de famille | 1 |
| Primary care doctor | 3 | | |
| Rural doctor | 2 | | |
| Family doctor | 1 | | |
| Generalist | 1 | | |
| Medical doctor | 1 | | |
| Medical practitioner | 1 | | |
| Primary health care doctor | 1 | | |

NB: there were multiple designations of doctors in some papers.

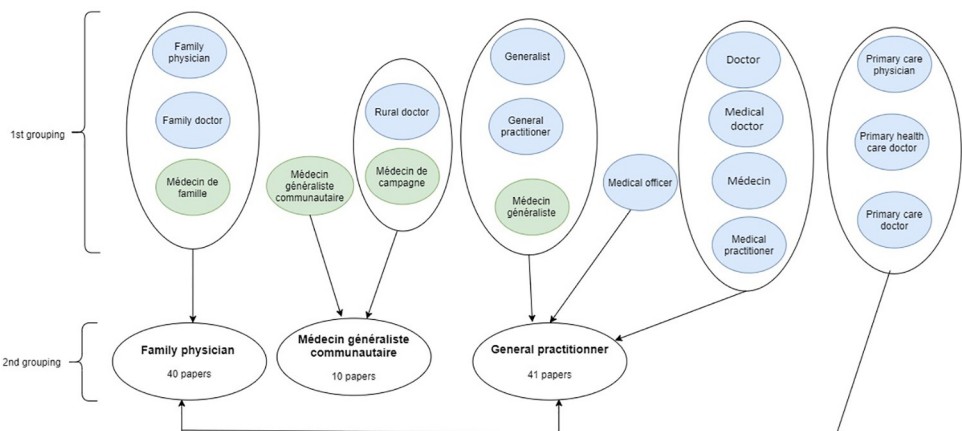

**Fig 2. Categorisation of primary care physicians.**

The category "family physician" (FP) corresponds to a type of PCPs with specific professional training in family medicine and primary care principles. FPs are mainly present in anglophone African countries (S3 Table), and 40 of the papers included in this review were related to this category.

"Médecin généraliste communautaire", "médecin de campagne" and "rural doctor" are put together in a second group. They refer to PCPs' whose practices were developed in francophone Africa under the impetus of the same French non-governmental organisation Santé Sud and are characterised by similar professional development. We use the label "médecin généraliste communautaire" (MGC) when referring to this category. The MGCs are only present in francophone African countries (S3 Table), and 10 papers were related to this category.

A third category grouped "general practitioner", "generalist", "medical officer" and "médecin généraliste". These names capture PCPs with no or unclear formal postgraduate training in delivering first-line care and whose professional development and identity are less pronounced and codified than the two former categories. We use the label "general practitioner" (GP) to designate this category.

The terms "doctor", "medical doctor", "medical practitioner" and "médecin" were used generically in seven papers to designate PCPs. We also allocated them to the GPs category because the contents of these papers clearly relate to doctors practising first-line general medicine.

The three names "primary care doctor", "primary care physician", and "PHC doctor" are rather broad terms. We do not believe they should constitute a distinct fourth category. Of the eight papers concerned, seven referred to PCPs in the GPs category and the other to PCPs in both the FPs category and the GPs category.

The GPs are found throughout SSA (S3 Table), and 41 of the papers included in this study were related to this group. However, it should be noted that, contrary to the papers relating to the other two categories, the information contained in these 41 documents on GPs is for the most part, fragmented and limited.

## Analysing PCPs practices in SSA

The data analysis yielded 13 main themes (Table 4). We regrouped them into four broader themes (called dimensions) based on the interrelationship between them.

The "professional identity" is the set of attributes, beliefs, values, motives, and experiences that people (and their entourages) use to define themselves in their professional work [33]. In the medical field, professional identity formation was identified as one of the key goals of

**Table 4. Main dimensions for analysing FLMDs' practices in SSA.**

| Themes | Dimension |
|---|---|
| • **Historical pathway**<br>• **Institutional status: public or private**<br>• **Postgraduate training**<br>• **Values explicitly put forward** | Professional identity |
| • **Policy guidance**<br>• **Integration into national health policies**<br>• **Integration into the local health system**<br>• **Resource allocation** | Governance |
| • **Roles assigned to the PCPs**<br>• **Actual activities performed and the setting in which these activities are performed**<br>• **Coordination of the PCPs' activities with the ones of other actors in the local health system** | Roles and activities |
| • **Impact on quality of primary care**<br>• **Impact on health outcomes and population health** | Outputs and outcomes |

medical education. It would indeed give the trainees the essentials values, skills and aspirations needed to fulfil their professional role [34]. Based on the above definitions of professional identity, we regrouped under this dimension, the postgraduate training received by the PCPs, if any, and the values in which their practices are embedded as reported in the analysed papers. Furthermore, the professional identity is constructed through dyadic interpersonal relationships (with trainers, patients, peers for instance.), group interactions and professional discourse [35]. We thus included within this dimension the PCPs' institutional status (public or private) and the historical pathway for the development of each category of PCPs, as these two themes may influence the PCPs' norms, believes and discourses.

Governance is a key function of health systems that entails policy guidance, regulation and coordination of different functions and actors, optimal allocation of resources, and ensuring accountability towards the population and all stakeholders [31]. For the governance of PCPs, we described the policies guiding their practices when they exist. We also analysed whether the national health policies integrate the PCPs among the resources needed to achieve the health system's goals (integration into national health policies). Integration into the local health system refers to the functional relationships of the PCPs with the local health authorities and their implication in the district health activities beyond their own activities. We finally examined which resources were allocated to PCPs.

In the "roles and activities" dimension, we reported the roles assigned to the PCPs and how they were defined. We then described the actual activities performed by the PCPs and the setting in which these activities are performed. Some PCPs work exclusively in first-line care, whereas others work both at first-line and hospital. Finally, we analysed the coordination of the PCPs' activities with the other health workers.

In the last dimension, we reported the available evidence on the outputs and outcomes of the PCPs' practices. As the main function of the first-line is to deliver primary care, we checked the impact of the PCPs practices on the quality of primary care, namely on accessibility, technical quality (for clinical processes), patient-centeredness, continuity, comprehensiveness and coordination. We then reported the existing evidence on the impact of these practices on health outcomes (for instance child mortality, maternal mortality, mother to child HIV transmission rate, etc.).

## Professional identity of PCPs

**Family physicians.** FPs have a long history in the RSA and Nigeria, where they existed before the 1980s [18, 36–38]. In other countries, their presence is much more recent [22, 39–45].

The FPs in SSA are promoted by academics, with the support of international institutions, such as the WONCA and Canadian and Belgian universities [21, 37, 38, 41, 42, 44, 46–48].

FPs work in both the private and public sectors [18, 22, 43, 49–51]. Most of the papers in this review reported experiences from the academic development of FPs in SSA, and a some of graduates are deployed into the district health system [39, 42, 50, 52, 53].

African FPs receive postgraduate training in family medicine for 2 to 4 years [36, 39, 43, 44, 54–58], and institutionalised postgraduate training programmes exist in a number of countries [39, 41, 43, 45, 46, 59–61]. In the RSA, there is a national consensus among the health authorities and academics regarding the core elements of the curriculum [46, 47, 57, 62]. This curriculum aims to provide FPs with a broad range of clinical skills, including basic medical and obstetric skills, but also more advanced skills (femoral vein puncture, vasectomy, etc.) [46, 57, 60, 63]. The other countries have similar curricula (with some adaptations) [40, 41, 44, 49, 55, 64]. The "Primary Care and Family Medicine Education Network" (PRIMAFAMED), created in 2008 and the collaborative projects that preceded it since 1997 [65], helped in this harmonisation of FPs' training in SSA [41, 46, 61]. After the postgraduate training, FPs are recognised as specialists [18, 39, 46, 50, 51]. This increases the interest of young doctors for family medicine as a life-long career [39, 60, 63].

The FPs' practices are meant to be based on values underlying primary care, such as a biopsychosocial approach to patient care [37, 41, 46, 48], community-oriented practice [62], universal access to comprehensive and quality care [36], and public service logic [39], among others. FPs strive to apply these values [39, 41, 46, 62], but their efforts are sometimes undermined by contextual factors, such as the hierarchical culture [66] or high workload [39, 67].

**Médecins généralistes communautaires.** The MGC programme was initiated in Mali in 1989 by the French NGO Santé Sud to support young doctors working in first-line health services in rural areas [68, 69]. The programme was progressively extended to Madagascar, Benin, and Guinée-Conakry [17, 70].

The MGCs work in the private sector and claim a not-for-profit private status [17, 18]. Before their installation, they receive theoretical and practical postgraduate training for 4 to 8 weeks [70, 71]. The training is provided by Santé Sud in collaboration with local universities and the MGCs' national associations [68, 70–72]. The training addresses the concept and values of the community-based general practice, the particularities of medical practices in isolated settings (routine preventive and curative care and the care of common emergencies), and health facility management [70–73]. A 2-year follow-up in the field by senior MGCs complements this initial training [70, 72, 73]. In Benin, the training entitles the doctors to a certificate in community-based general practice [70, 74].

Like FPs, the MGCs' practices should also be based on primary care [17, 69–71, 73]. Evaluations in Mali and Benin have found that working for rural and underserved communities provides an inner motivation to the MGCs [69, 73]. However, many MGCs wish to obtain a specialisation or move to another job after a few years [69, 73].

**General practitioners.** The GPs, by the categorisation adopted in this review, are medical doctors without a formal (or with unclear) postgraduate training addressing the specificities of the medical practice at primary care level [62, 75–77]. Although some countries have had GPs at the first-line for decades, this started to expand in the late 1990s, following the private sector's development [6, 18], where many GPs work [66, 76, 78–80]. Several papers reported that the GPs constitute the majority of the PCPs in their settings [62, 76, 81, 82].

For a number of the private GPs, it is a personal choice and decision to settle in first-line care [83]. However, in the public sector, GPs are placed at first-line facilities by the Ministry of Health to introduce an intermediary level between the first-line managed by non-physicians and the district hospitals [50, 84, 85]. We also found examples of deliberate efforts from the

government to provide PCPs to underserved communities. This is the case of Cuban-trained doctors in the RSA, for instance [86].

The GPs' practices appear to be poorly codified, and we could not find an explicit definition of values for this group [62, 78]. Some authors have reported that the practices of GPs in the private sector are largely business-oriented [25, 75]. However, we came across some examples in which private GPs opted for publicly oriented service delivery, especially when exposed to the values underlying PHC [87, 88].

Finally, in many cases, GPs do not consider their first-line work as a life-long career. After some time, they go for specialisation or other types of work [70, 89].

## Governance of PCPs' practices in SSA

**Family physicians.**   In the RSA, the FPs' practices are guided by a set of nationally agreed-upon roles validated by the health authorities and academics [37, 46, 60, 62, 63, 67, 82, 90]. In this country, FPs are acknowledged in several policy documents [46, 58, 60], although it appears that these documents still need to fully apprehend and integrate the FPs' roles [51, 58]. In other countries, we did not find formal policy guidance for FPs, and the literature indicates insufficient guidance in some countries [41, 45, 49]. Besides, FPs are not formally integrated into the health policies [19, 38, 48, 49, 91], in many countries. In the best cases, such as Ethiopia or Kenya, they are mentioned in health policies as contributors to PHC [42, 45, 61]. Conversely, in Rwanda, the training of FPs was cancelled in 2011 following a change at the Ministry of Health [38].

The FPs community strives to properly integrate FPs into district health systems in SSA [39, 41, 50, 63]. However, in many cases, there is not yet a clear understanding of their role in the (local) health systems [19, 45, 55, 82, 85]. Policymakers and district managers often do not know how best to use the FPs [33, 36, 46, 82], and there is sometimes an inadequate positioning of FPs (as hospital managers or consultants at tertiary hospitals for instance) [58, 82, 85, 91]. Moreover, the coordination with other actors in the local health system is suboptimal as we will see in the section on roles and activities.

In terms of resources, training and hiring a sufficient number of FPs is challenging [22, 43, 60] and much more expensive than for non-physicians, or even other PCPs [18, 22, 60, 90]. Some studies indicate an insufficient number of FPs, indeed [45, 49–51]. Family medicine initiatives in Africa have gained support from the PRIMAFAMED network and other collaborations [41, 46, 61]. However, studies reported that the support from the national health authorities is still insufficient, in the few countries where it exists [22, 38, 39, 41].

**Médecins généralistes communautaires.**   MGCs' practices are guided by a charter through which the MGCs commit to a publicly oriented practice and an alignment with the national health policy. This charter is signed between the MGCs, their professional association, the community representatives (municipalities and/or local associations), and the local health authorities [18, 69, 70, 74]. The charter defines each party's rights and obligations and the activities to be performed by the MGCs [17, 70]. The MGCs professional associations, the Association des Médecins de Campagne in Mali and Association des Médecins Généralistes Communautaires in Benin, ensure compliance of MGCs with their commitments [17, 18, 69–71, 73].

However, the MGCs are not officially integrated into national health policies [70, 73]. At the local level, the MGCs receive in-service training/supervision from the health district managers or the hospital physicians [69]. They also participate in hospital duties or other health activities within the health district [17, 69, 73]. However, the degree of this integration into local health systems varies [69, 70]. In Mali, most MGCs work in community health centres,

which are fully integrated into the district health map even if their management has been devolved to local health associations. The State also contributes to their salaries [17, 69, 72, 73]. In Benin, the MGCs work mainly on a self-employment basis, and their relationships with the district health authorities range from full support to indifference, or even opposition [70].

The MGCs receive support from their professional associations and Santé Sud for the initial and in-service training [18, 69, 70, 72], for their installation [17, 72], and for facilitating their relationships with the health authorities [17, 18, 69, 73]. Sometimes, the local governments help the MGCs with the health centre's premises or equipment [18, 69, 70]. However, the MGCs' initiative is still too dependent on funding from Santé Sud, with little local financing and support from national health authorities [68, 69]. Consequently, there are very few MGCs [69, 70].

**General practitioners.**   The public GPs work under the general rules of public service governance [89, 92, 93], but this is not specific to the medical practice at the primary care level. In the private sector, it is difficult to precisely describe the governance rules applying to the GPs. In the RSA, there have been efforts to guide private GP practices through the roles adopted for the FPs [62, 82], but the GPs appears to be rarely involved in defining these roles, and it seems that they do not usually adhere to them [37, 58]. Some measures have also been adopted in many countries for guiding PCPs practices, such as contracting with private GPs within the frame of national health insurance [82, 94] or broader regulation measures targeting the private sector [87]. However, these measures seem scarce and fragmented [83, 87, 95]. Additionally, they are indications that key aspects (ensuring the quality of care or controlling the costs for the patients, for instance) are not always regulated [87, 94, 95].

Regarding the integration into national health policies, many countries' policy documents mention that some health centres should have a GP [19, 89, 96] but do not go further. At the local health systems level, the collaboration between private GPs and health authorities is not always going smoothly. The latter complain that private GPs do not usually report their activities or comply with the health policies [78, 97, 97], and the GPs often complain about little support and not being involved in public health programmes [94, 98, 99].

Many papers reported that private GPs do not always get the appropriate information regarding national policies and guidelines [57, 66, 92, 95, 97, 100] and lack adequate support to deal with the difficulties encountered [79, 89, 100–102]. In many cases, private GPs have to rely on private pharmaceutical or insurance companies for training opportunities [46, 87], and their activities are largely financed by the patients' payments [87, 99].

## Roles and activities

**Family physicians.**   The FPs support and lead healthcare teams within the health district through six roles: (i) provision of clinical care, (ii) consultations for patients referred by other primary care team members, (iii) mentoring and training other clinical staff, (iv) leading clinical governance activities within the health district, (v) supervising undergraduate or postgraduate students, and (vi) championing community-oriented primary care [47, 50, 53, 57, 58]. These roles were developed in the RSA, but other English-speaking African countries have adopted similar roles thanks to several North-South and South-South collaborations such as the PRIMAFAMED network or partnerships between African universities for implementing family medicine in several countries [39, 41, 46, 49, 103].

In many cases, first-contact care is not a principal activity of FPs [38, 60, 67, 83, 88]. When they are posted at first-line facilities, they manage complex cases referred to them by the nurses, supervise the staff, perform quality improvement activities, and organise community-based activities [36, 39, 60, 67]. They may also be based at the hospital level [39, 43, 50, 58] and provide outreach support to primary care teams at the first-line [39, 60, 62, 67]. At the hospital,

the FPs provide a wide range of services, including surgical and anaesthetic procedures [36, 44, 57]. These hospital duties may divert the FPs from their primary care responsibilities, as reported in many studies [21, 39, 41, 60, 67, 83, 90]. FPs can also have managerial positions at district hospitals or district management teams [22, 60, 101].

Due to this wide range of activities, FPs have sometimes overlapping roles with other actors, particularly other specialists and districts managers. This leads sometimes to tension and conflicts [36, 52, 58, 60, 61, 104].

**Médecins généralistes communautaires.** The roles assigned to MGCs are to (i) provide first-contact care, (ii) organise healthcare activities and manage the health facility, and (iii) engage in local development by advising the local government on health and hygiene matters [17, 69–71]. Accordingly, they organise community-based preventive activities, provide curative care, supervise their team, plan and monitor health activities, and engage with local government and community leaders [17, 73].

MGCs quasi-exclusively provide first-contact care [18, 69]. In Mali, Madagascar, Benin, and Guinée-Conakry, they offer the minimum healthcare package defined by the government, including vaccination, family planning, antenatal care, deliveries, and all-round curative care for common diseases [17, 69, 70]. Studies also reported that they go beyond this minimum package by providing services such as dystocic deliveries, minor surgery, home visits, and care for chronic diseases [69, 70, 74, 105].

The roles and activities of the MGCs can conflict with the roles of non-physician clinicians providing first-contact care. In Mali, for instance, nurse-practitioners, previously heading the health centres and providing the first-contact care, were replaced by MGCs. This obviously led to tension [69]. Also, health authorities consider some MGCs' activities (e.g., inguinal hernia surgery) as going beyond what is allowed at the first-line by the health policies [70, 72, 73].

**General practitioners.** We did not find a clear description of the role of GPs, and the scope of their activities varies considerably, even in similar contexts [18, 62, 66]. Private GPs offer first-contact care [62, 87]. In contrast, public GPs tend to work in support of nurse-practitioners through supervision [19, 66, 83, 89, 96] and management of complex clinical cases [50, 84, 85, 96]. These public GPs can perform preventive activities and often have managerial tasks [78, 89].

Both public and private GPs offer care that goes beyond the minimum healthcare package that is classically offered at the first-line, for instance, care for high-risk pregnancies [106], depression [54], pleurisy [82], and chronic diseases [100], among others. First-line public GPs may also have hospital duties by which they provide emergency care, and even major surgery [82, 88, 89]. On the other hand, hospital-based GPs can be officially mandated to provide outreach support to primary care teams [19, 62, 78, 81, 87, 96, 106].

However, the expansion of first-line activities by GPs is not without problems. Indeed, they may engage, sometimes quite problematically, in controversial activities, such as experimentation with a herbal HIV treatment [87] or major surgery at the level of first-line health services [78]. Public GPs are sometimes accused of neglecting clinical work in favour of administrative tasks [89]. Besides, the public GPs may perform consultations at private first-line facilities in a clandestine manner [78, 87].

Finally, conflicts can emerge between GPs and other health workers due to overlapping roles. For instance, nurses and GPs sometimes compete for patients in the private sector [17, 25]; and power struggles can emerge between them in the public sector [85, 88].

## Output and outcomes

**Family physicians.** Several studies reported that FPs improve primary care. There is an impetus to position them in places where there are most needed, including in rural and/or

underserved areas [22, 45, 107], which increases the accessibility of care. Additionally, in many settings, FPs helped manage complex cases and chronic diseases within the health district [19, 39, 58, 104], thus saving costs for the health system and patients [22]. Districts managers and other health workers reported that FPs improve clinical processes by training/mentoring other primary care providers and initiating quality assurance activities [39, 58, 60, 67, 90, 104]. Stakeholders also reported a positive impact on health outcomes such as decreased mother-to-child HIV transmission through better antenatal care [60]. An observational study corroborated this positive impact on accessibility and clinical processes. This study found that the presence of FPs was associated with better availability of essential services and fewer modifiable factors contributing to pediatric deaths [107].

However, some studies found shortcomings concerning communication with the patients [66, 108], coordination of care [107], continuity of care [88, 107], comprehensiveness [21], and community orientation [67, 104]. Furthermore, we found only a few observational studies that have evaluated the impact of FPs on health outcomes (such as child mortality, maternal mortality or mother to child HIV transmission rate). These studies were limited to the RSA. They could not demonstrate a significant impact of the FPs on health outcomes [50, 53, 107, 109], despite the positive impacts reported on clinical processes and accessibility. This was attributed to the limited number of FPs [50, 53, 107, 109].

**Médecins généralistes communautaires.** Evaluations found that MGCs improved the technical quality of first-line care through better clinical procedures and regular staff supervision [69, 70]. An increase in service utilisation was also reported [17, 73]. Some studies reported that MGCs improved access to essential medicines [70], and access to chronic diseases care and other services that were rarely offered at first-line facilities [18, 69, 70, 73, 105]. They improved the continuity and the coordination of care through, for instance, follow-up of patients via home visits [17, 18, 69, 73], better referral procedures and better communication between the first-line and hospitals [69, 70]. MGCs also try to improve financial access to care by applying lower costs than the classical private structures [69, 70] and promoting community-based health insurance [17, 18, 69, 70].

However, their presence may entail increased costs for the health facility and, consequently, for the patients [69]. This could imply that MGCs, who mainly work in rural areas, would preferably settle in areas where the community can afford this higher cost [69].

Regarding the impact of MGCs on health outcomes, one study conducted in Mali has mentioned a contribution to better health outcomes; for example, an 80% decrease of epileptic seizures among MGC patients [73]. However, there is still not sufficient evidence to confirm this impact [69, 72].

**General practitioners.** Some studies found that GPs can effectively provide care at first-line facilities for health problems that are too complex to manage with simple algorithms, such as HIV and other chronic diseases [82, 100]. However, many studies reported that a significant proportion of GPs (both public and private) fail to apply national or international standards for the care of common diseases [56, 75, 80, 92, 93, 95, 97, 110–113]. Some papers also reported that GPs can improve the access to care in underserved areas [86, 98, 100]. However, access to the GPs' services in the public sector can be limited by a long waiting time [25, 87, 98] or the fact that the nurses may not refer the patient in time [81]. The private GPs offer an alternative for this little availability of the public GPs [94, 98], and they improve the availability of health services that are not offered in the public sector [25, 81, 87]. However, several studies have reported that private GP services are mainly accessible to people who can afford it [18, 25, 75, 81, 87, 95, 99, 113] or those in urban areas [18, 79, 87].

There are also shortcomings regarding patient-centredness, such as inadequate communication with the patients [92, 95, 96] or stigmatisation of mental health patients [54]. We also

**Table 5. Main characteristics and key issues related to PCPs' practices in SSA.**

|  | Family physicians | Médecins généralistes communautaires | General practitioners |
|---|---|---|---|
| Professional identity | • Promoted by academics<br>• Mainly in anglophone countries<br>• Extensive postgraduate training of 2 to 4 years with a specialist status<br>• Both public and private<br>Embedded in the values of primary care<br>• Few | • Promoted by the French NGO Santé Sud<br>• Only francophone countries<br>• Postgraduate training of 4 to 8 weeks with 2-year follow-up<br>• Mainly private<br>• Embedded in the values of primary care<br>• Very few | • Mainly spontaneous and unstructured development<br>• Throughout Africa<br>• Usually no postgraduate training specific to primary care<br>• Both public and private, but mainly private<br>• Little information on the values underlying practices<br>• Reported to be the majority of PCPs in many settings |
| Governance | • Practices guided by a set of nationally agreed-upon roles in the RSA; no formal policy guidance found for other countries<br>• Still several issues despite efforts to integrate FPs into local health systems and national policies (lack of a proper understanding of their role, inadequate positioning, etc.).<br>• Difficult to train and hire a sufficient number due to high costs<br>• Insufficient support from the national health authorities | • Practices guided by a charter signed between them, their association, and local authorities<br>• Not officially integrated into national health policies; variable integration into the local health system,<br>• Training and installing largely relies on funding from a French NGO<br>• Insufficient support from the national health authorities | • No formal and comprehensive policy guidance found and issues with the regulation of the private practice<br>• Many issues regarding the integration into the local health system and national policies-<br>• High reliance on private funds (patients, private companies)<br>• Insufficient support from the national health authorities |
| Roles and activities | • Six roles: (i) care provider, (ii) consultant members, (iii) mentor and trainer of the clinical staff, (iv) leader of clinical governance, (v) supervisor of students, and (vi) champion of community-oriented care<br>• First-contact care seems not to be the main activity, rather function as referral support to other primary care workers<br>• Can also provide hospital-based clinical care, with specialised clinical procedures<br>• Overlapping roles with other actors (specialists ++) | • Three roles: (i) care provider, (ii) manager of the health facility, and (iii) advocate for local development<br>• Quasi-exclusively provide first-contact care<br>• Offer the minimum healthcare package and more (e.g., dystocic delivery)<br>• Overlapping roles with non-physicians and some first-line activities (e.g., surgery) conflict with national policies | • No clear description of their role found<br>• Considerable variation in the scope of activities (sometimes provide first-contact care, sometimes function as referral support and sometimes have a mixed practice)<br>• Offer the minimum healthcare package and more<br>• Overlapping roles and competition with non-physicians |
| Output and outcomes | • Improve the technical quality of care<br>• Improve the geographic access to care for complex cases and chronic diseases, saving costs for the system and patients<br>• Shortcomings in terms of continuity of care, comprehensiveness, and community-orientation<br>• Indication of positive impact on clinical processes but more evidence needed on health outcomes | • Improve the technical quality of first-line care<br>• Improve access to medicines, access to care for chronic diseases and other services, continuity and coordination<br>• Evidence lacking on their impact on health outcomes | • Can effectively provide care for complex cases and chronic diseases but the technical quality of care may be problematic<br>• Shortcomings regarding patient-centredness, comprehensiveness, access to services and continuity<br>• Evidence lacking on their impact on health outcomes |

found issues related to comprehensiveness as the GPs (especially the private ones) may focus on curative care [76, 83] or some specific services [87]. The continuity of care is not also always optimal [75, 87, 89, 98]. Nevertheless, it should be noted that, given the diversity observed in the GPs' activities, these issues are not necessarily found among all them.

We did not find studies assessing the impact of GPs on health system outcomes.

Table 5 provides a summary and the key issues related to PCPs' practices in SSA.

## Discussion

### Heterogeneity in the PCPs' practices in SSA

This review distinguished three groups of PCPs in SSA, and there is a great diversity in their professional identity, the governance of their practices and their roles and activities. FPs and MGCs receive specific training for the delivery of primary care, and the values and roles underlying their practice are explicitly stated. Both groups are actively promoted by academics

or NGOs [26, 50, 53, 68, 69]. However, their training differs significantly (a few weeks for MGCs and up to 4 years for FPs), as well as their activities. As for the GPs, they do not benefit, in many contexts, from formal postgraduate training, which would have prepared them for the specificities of the primary care-level work [56, 62, 93, 101, 114, 115]. The roles, the activities and values underlying the GPs practices seem relatively less codified, compared to the FPs and the MGCs.

There is also diversity within each category. For example, in the RSA, FPs are trained for 4 years and work at all levels of care (first-line, second-line and even tertiary level in some instances [22, 101], while in Sudan they are trained for 2 years and are rather positioned in community-based health centres [22, 64]. The variation is especially great among GPs, with differences appearing between the public and the private GPs, calling for more empirical research to verify and specify these differences. Additionally, the findings of this review suggest that the practices of private GPs are quite heterogeneous, making it difficult to predict their contribution to primary care.

The diversity in the PCPs practices in SSA may be partly linked to the various pathways in their evolution. For example, FPs have benefited from a long period of maturation during which issues like professionalism, core principles of training, the necessity to be rooted in the values of primary care and to be responsive to the needs of the population, have been long discussed and conceptualised, especially in the RSA [37, 48, 57, 104, 116]. Therefore, it is not surprising that their professional identity and roles appear to be relatively more codified than for the other two groups. The MGCs' initiative was developed, under the impetus of a French NGO, to enable rural African populations to benefit from the care provided by a physician. Therefore, it is understandable that these doctors are found almost exclusively in francophone Africa and that they provide first-contact care (somewhat like the family physicians in France). As for the PCPs classified in the GPs category in this review, especially those in the private sector, their installation at the first line seems rather spontaneous and driven by circumstances such as the need of professional integration in an environment where access to the civil service and specialised training remains limited and where market opportunities are opening up with growing urbanisation and the development of a middle class within the population.

The medical practices at the primary care level should be adapted to the community's needs, resources, and local contexts [22, 89, 117]. Therefore, efforts to improve primary care in SSA should consider this great diversity in the practices of PCPs as each category implies different opportunities and challenges. Unfortunately, the current scientific debate on physicians' contribution to primary care in SSA almost exclusively focuses on FPs [21, 22, 38, 117–119], with a lack of comprehensive information on the two other categories. The information gathered for the GPs is especially fragmented. Also, the literature from the RSA largely dominates. These research gaps could be attributed to publication bias. There may be more research and publications on FPs, and to a lesser extent on MGCs, because there are actively promoted by academics and NGOs (more motivated to publish their work), which is not the case for many GPs. Similarly, the RSA is reported to be the highest producer of research outputs in SSA [120, 121], and researches and experiences from the other countries may not be sufficiently published and indexed.

## Several issues related to the governance of PCPs

This review highlighted many issues with the integration of PCPs into local health systems and national health policies. Even if some policy guidance exists for the FPs, some papers reported a lack of a proper understanding of their roles and an inadequate positioning within the local

health system [45, 49, 58]. For all three categories, the activities of PCPs may conflict with those of other cadres in the health system. Our findings also suggest regulatory issues, especially for the practices of the private GPs [87, 96], and gaps in the health policies regarding the PCPs [38, 70, 73].

These governance issues may be explained by the fact that, in some African countries, there is not yet a clear vision or consensus on how PCPs could contribute to the health systems' goals and how they could interrelate with other health system actors [19, 36, 39, 50, 61, 70, 91]. For instance, a study on the perception of FPs among African leaders reported that many policymakers have not yet conceptualised the contribution FPs can make on their health systems [91]. In addition, health authorities may lack the necessary resources (funds and management tools) to fully integrate PCPs into health policies [39, 104].

Inadequate governance of PCPs may threaten the performance of primary care. Insufficient policy guidance and poor regulation mays lead to sub-standard clinical practices, uncontrolled increases in costs, and a focus on individual and curative care, as it was actually shown in some studies [18, 78, 87]. Moreover, in many cases, PCPs (mainly the private GPs, but also some MGCs and FPs), rely on private resources and input (out-of-pocket payments, subsidies and training by private-for-profit organisations, for example). This may greatly influence the activities performed [66] and the population they serve [17, 18, 87, 99], with a threat to accessibility and other primary care aspects. Furthermore, lack of recognition, insufficient support from the health authorities, and little professional development opportunities may be demotivating factors for PCPs and lead to their migration to foreign countries or non-clinical fields (health management for instance). Indeed, strain and emotion (partly due to poor working conditions), absence of adequate professional support and desire for professional prestige and respect are among the common drivers of health workers' migration [122].

Therefore, for PCPs to effectively contribute to primary care in SSA, sound governance of their practices will be necessary. As recommended in the governance and leadership literature [123, 124], countries will need to clarify the roles expected for PCPs, provide the necessary resources, clarify the ways the outcomes will be measured, provide regular feedback to the PCPs, and clarify the consequences if expectations are (not) met. The values in which the PCPs' practices must be embedded should be made explicit [31], and their interrelations with the other primary care workers should be considered.

## Contribution of PCPs to primary care in SSA

All three groups expand the range of available first-line services, and many papers stated that this improves the geographic access to care and reduces referrals. The literature also reported that the FPs, MGCs, and some GPs improve the technical quality of care.

However, this review shows that, in many cases, the PCPs fail to address some key principles of good primary care (person-centeredness, continuity, comprehensiveness, coordination and community-orientation [5, 125, 126]. For example, the hospital duties of the FPs and the contextual constraints may sometimes divert them from properly addressing these features of good primary care, and many PHC experts have raised concerns about it [22, 118, 127]. There were also concerns regarding the accessibility [18, 25, 75, 81, 87, 95, 99, 113], the comprehensiveness [76, 83, 87] and the continuity [75, 87, 89, 99] of the services offered by the GPs. Furthermore, if most of the paper reviewed on the MGCs' practices reported good compliance with primary care principles [17, 22, 70, 73], more empirical evidence is needed before drawing conclusions [69].

The mitigated performance regarding primary care specificities may be explained by the fact that stakeholders' main expectation for the PCPs in SSA is an improvement of the

technical quality of primary care services [18, 36, 73]. African FPs struggle to be recognised as "specialists" or "experts" [39, 41, 46] and, therefore, may be pushed to provide well-valued technical procedures [60, 118, 128]. The GPs try to show their "superiority" compared to non-physicians by performing surgery or more tests [17, 78]. Furthermore, many stakeholders argue that it is more efficient to use FPs, and even public GPs, in district hospitals given their limited number and the specialist shortage in these hospitals [22, 38, 50, 89].

However, we must be careful that the need for recognition and resource constraints do not take us away from primary care's essence and reinforce our health systems' current biomedical and hospital bias [6, 18, 129]. This again calls for the need for proper governance of PCPs. Some studies have shown that effective management [98, 99] and judicious task-sharing with non-physicians can mitigate the cost of permanently having doctors at the primary care level [98, 130]. Examples from Costa Rica and Brazil show that positioning doctors within multidisciplinary primary care teams is feasible and yields good results [8, 118]. Therefore, instead of focusing on task-shifting [115] or trying to replace non-physicians with doctors for first-contact care [74], African countries could consider task-sharing and teamwork as a way to use the scarce human resources more efficiently. A few papers already proposed this renewed vision [5, 7]. The roles assigned to FPs and public GPs tend towards this vision, but very little evidence indicates such teamwork in SSA [8].

Finally, the evidence on the impact of all three categories of PCPs on the quality of primary care is still insufficient and largely based on stakeholders' perceptions. This calls for more research in this area, as it was also concluded by a recent scoping review on FPs in SSA [22]. Moreover, the fact that the limited number of FPs make it difficult to demonstrate their impact suggests the need to adopt more appropriate research methods for assessing this impact.

## Strengths and limitations

This scoping review followed a rigorous methodology which supports an extensive and systematic scan on the available literature on PCPs in SSA. It complemented the previous evidence by going beyond the FPs and English-Speaking countries. It also used a health system perspective to analyse PCPs practices, which helped identify the main issues related to their contribution to primary care.

The first limitation of this study is that we did not appraise the quality of the paper included. This is not indeed the principal objective in scoping reviews, especially in this case were our objective was to map the key issues related to the PCs' practices. However, the conclusions of this study still need to be confirmed with good empirical studies.

Secondly, we could not include the government policy documents on the deployment of PCPs because it was difficult to get them for all SSA countries without a primary data collection. Also, for the grey literature, we searched on the websites of the WONCA and the NGO Santé Sud and included papers recommended by experts in the field of PHC. Although these sources were purposively selected, we might have missed some information on the practices of PCPs. Likewise, by excluding the doctors working exclusively in hospital, we might have missed research about PCPs working in rural districts or mission hospitals serving the population through outreach activities rather than a formalised network of other PHC facilities.

Finally, even if we went beyond the English literature by including papers written in French, we may have still missed important information on the PCPs' practices in SSA countries with other languages (such as Portuguese, Spanish, etc.).

We nevertheless did our best to moderate the impact of these limitations by conducting a careful review of the papers, including the screening of the introduction part and careful screening of their reference lists.

## Conclusions

This scoping review sheds light on the main characteristics of PCPs working at the first line of health care delivery in SSA. We distinguished three categories among these PCPs: the family physicians, the "Médecins Généralistes Communautaires" and the GPs. We analyse the functioning of each category along four dimensions that emerged from the data analysis. This analysis pointed to several challenges. The governance issues especially may threaten the effective contribution of PCPs to sound primary care in SSA. These issues could be handled with coherent governance arrangements, grounded in contextual evidence. However, important research gaps remain concerning the precise nature and performance of the PCPs activities, and the most appropriate governance modalities to put in place. These gaps are more pronounced for countries other than the RSA and PCPs classified in the category of GPs.

More studies will help to address these knowledge gaps, building on the interesting initiatives reported in the literature. These studies could also pay particular attention to the role of PCPs in addressing the emerging challenges that Africa and the world are currently facing, including COVID-19 and climate changes.

## Postscript on the positionality of authors

To ensure an analysis that considers the African context and its particularities, we involved in this work authors and contributors from several African countries and with various backgrounds. We discussed our findings with several African primary health care professionals. This process helped us integrate perspectives from many countries (see the paragraph "Stakeholders' consultation" in the methodology).

Our paper promotes African authors as they were at the forefront of data collection, analysis, and scientific writing. The non-African authors brought solid scientific experience for primary health care research, and excellent knowledge of the African context. However, the study was entirely led by the first author, an African female researcher. This contributes to building research capacity in Africa as well as a better credit to African authors, which are key steps towards decolonizing authorship in global health research [131, 132].

The detailed profiles of the authors are presented below:

- KB is a public health doctor and a General Practitioner. She worked for ten years (2008 to 2018) as a primary care physician at the first-line of healthcare delivery in Benin. She is currently performing a PhD research that focuses on the analysis and improvement of the practices of primary care physicians in Benin.

- JDLP is a professor of family medicine at the University KU Leuven in Belgium. He is a family physician for decades, and he is the leader of a writing group on mental health in Primary Care within the European Forum for Primary Care (EFPC). In his unit, several PhD programs are conducted in low-and-middle-income countries.

- JK and SB are Congolese physicians with a very good knowledge of their country's local health systems' organization. SB is a district medical officer, and he has good insights into the practices of primary care physicians in his health district. He recently conducted a study on primary care physicians in Kisangani, DR Congo. This study was recently published in French in the African Journal of Primary Health Care and Family Medicine.

- JPD is a public health doctor and health policy and systems researcher from Benin. He has a good command of several research methodologies, including the scoping review methodology.

- EW is a registered nurse from Kenya, a health systems researcher and a quality of care change agent. She has good experience of the health care organization within health districts in Kenya and Uganda. She is currently writing up and finalizing her PhD thesis on understanding the challenges and opportunities in implementing patient-centred primary health care in Uganda.

- LA is a Belgian clinical doctor, a lecturer and a post-doctoral researcher at the Institute of Tropical Medicine in Antwerp, Belgium. He is a qualified Family Physician who worked for several years in a primary care practice. He worked for more than ten years in Africa, mainly in Zimbabwe but also in Kenya, DR Congo and Ethiopia. In Zimbabwe, he held several positions at primary, secondary and tertiary health care levels as a public health officer and district medical officer.

- MZ is a professor of internal medicine at the University of Abomey Calavi in Benin. He has been teaching undergraduate medical students for more than twenty years. He is also involved in the training of other primary care clinicians, and he has been the deputy director of Benin's national nursing school for three years. Besides his teaching and clinical activities, he contributes to several initiatives for primary care improvement in Benin.

- BC is a professor of public health at the Institute of Tropical Medicine Antwerp. He has over 30 years of experience in the organization and management of local health systems and primary health care (including family medicine) in Belgium and a wide range of African and Asian countries. He is currently coordinating health systems strengthening projects in the Democratic Republic of Congo, Guinea (Conakry), India, Mauritania and Uganda.

## Supporting information

**S1 Text. Protocol of the scoping review on first-line medical doctors in sub-Saharan Africa.**
(PDF)

**S2 Text. PRISMA extension for scoping review checklist.**
(PDF)

**S1 Table. Search strategy on MEDLINE.**
(DOCX)

**S2 Table. Data extraction form with the data extracted from the studies included in the scoping review.**
(XLSX)

**S3 Table. List of papers included.**
(DOCX)

## Acknowledgments

We acknowledge guidance and input from Dr Zakaria Belrhiti (Morocco) and Mr Dirk Schoonbaert (Belgium) in the development of the review protocol and the literature search. We are also grateful to Dr Mohamed Ali Ag and Dr Delphin Kolie for commenting on the findings of this review based on their experience as former primary care physicians and researchers in Mali and Guinée, respectively. We finally thank the MPH and PhD students from the Institute of Tropical Medicine, our colleagues from the Institute of Tropical Medicine and our colleagues from the *Centre de Recherche en Reproduction Humaine et en Démographie* for useful discussions and advice.

## Author Contributions

**Conceptualization:** Kéfilath Bello, Jan De Lepeleire, Bart Criel.

**Data curation:** Kéfilath Bello, Jeff Kabinda M., Samuel Bosongo.

**Formal analysis:** Kéfilath Bello, Jan De Lepeleire, Jeff Kabinda M., Samuel Bosongo, Evelyn Waweru, Bart Criel.

**Funding acquisition:** Kéfilath Bello, Jan De Lepeleire, Bart Criel.

**Methodology:** Kéfilath Bello, Jan De Lepeleire, Jeff Kabinda M., Jean-Paul Dossou, Ludwig Apers, Marcel Zannou, Bart Criel.

**Project administration:** Kéfilath Bello.

**Resources:** Bart Criel.

**Supervision:** Jan De Lepeleire, Marcel Zannou, Bart Criel.

**Validation:** Jan De Lepeleire, Evelyn Waweru, Bart Criel.

**Visualization:** Kéfilath Bello.

**Writing – original draft:** Kéfilath Bello.

**Writing – review & editing:** Kéfilath Bello, Jan De Lepeleire, Jeff Kabinda M., Samuel Bosongo, Jean-Paul Dossou, Evelyn Waweru, Ludwig Apers, Marcel Zannou, Bart Criel.

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
