## [Decision Letter · Decision Letter 0]

6 Nov 2020

PONE-D-20-25494

The expanding movement of medical doctors operating at the first-line of healthcare delivery systems in sub-Saharan Africa: A scoping literature review

PLOS ONE

Dear Dr. Bello,

Thank you for submitting your manuscript to PLOS ONE. After careful consideration, we feel that it has merit but does not fully meet PLOS ONE’s publication criteria as it currently stands. Therefore, we invite you to submit a revised version of the manuscript that addresses the points raised during the review process.

We look forward to receiving your revised manuscript.

Kind regards,

Virginia E. M. Zweigenthal

Academic Editor

PLOS ONE

Journal Requirements:

2. Please amend your list of authors on the manuscript to ensure that each author is linked to an affiliation. Authors’ affiliations should reflect the institution where the work was done (if authors moved subsequently, you can also list the new affiliation stating “current affiliation:….” as necessary).

Additional Editor Comments:

Dear Author,

Thanks you for your submission. Our apologies for the time it took for this review.

Please note the comments from the reviewers and you are invited to revise your submission based on their comments.

Many thanks,

Dr Virginia Zweigenthal

Reviewers' comments:

Reviewer's Responses to Questions

**Comments to the Author**

1. Is the manuscript technically sound, and do the data support the conclusions?

Reviewer #1: Partly

Reviewer #2: Partly

2. Has the statistical analysis been performed appropriately and rigorously? 

Reviewer #1: N/A

Reviewer #2: N/A

3. Have the authors made all data underlying the findings in their manuscript fully available?

Reviewer #1: Yes

Reviewer #2: No

4. Is the manuscript presented in an intelligible fashion and written in standard English?

Reviewer #1: Yes

Reviewer #2: No

5. Review Comments to the Author

Reviewer #1: Thank you for the opportunity to review this manuscript. This scoping review examines the practice of first-line medical doctors (FLMDs) in sub-Saharan Africa. The novel aspect of this scoping review is that, as opposed to other reviews, this paper also looks at the roles of FLMDs in francophone countries, as well as non-specialist primary care doctors.

The following comments are made to assist the authors enhance the manuscript:

1. The term "FLMDs" sits uncomfortably - not sure if this may be due to translation? The accepted term is primary care doctors or primary care physicians (general/generalist doctors providing first-contact care at the primary level of care in the health system). The term family doctor is used by WONCA. This reviewer would strongly recommend that the authors consider changing to one of these terms, as "FLMDs" makes it difficult to relate to other research on the supply of primary care doctors or the primary care workforce (human resources for health). If the authors strongly feel that they would like to stick to the new term, they will have to make a stronger case, as the current text is unconvincing.

2. On page 15, line 211, it is stated that the PRIMAFAMED network was created in 2017. This is factually incorrect. The origins of this network may be traced back to 1997. The name Primafamed was used for the first time in 2008. See editorial by founder Prof Jan de Maeseneer: https://phcfm.org/index.php/phcfm/article/view/1603/2247.

3. On page 17, lines 250 onward, you might also wish to look at the national position paper drafted for the South African health department: https://scholar.sun.ac.za/handle/10019.1/99785; and a recent paper in the South African Health Review: https://www.hst.org.za/publications/South%20African%20Health%20Reviews/Chap%204%20Family%20Physicians.pdf. You will find that even though it appears that FPs are integrated in health and human resources policies, there is still a fair degree of ambiguity around the roles and scope of practice of FPs.

4. On page 20, lines 312 - 314: whilst the Primafamed network has played a big role, it is incorrect to attribute all the expressions of family medicine in Africa to this network. There are many additional North-South and South-South influences over several decades, especially Nigeria, Uganda, Kenya and others.

5. It has to be stated that the term general practitioner also carries a fair degree of ambiguity and confusion. In the international context, the terms general practitioners and family physicians/doctors are often used interchangeable for primary care doctors with 2 - 4 years of postgraduate training (see the Besrour series and the WONCA publications). Furthermore, in South Africa general practitioners refer to primary care doctors without postgraduate training who work in first level of care in the private healthcare sector, whereas medical officers refer to primary care doctors without postgraduate training who work across the levels of care in the public healthcare sector.

6. On page 27 in line 415, this is one example where the authors state that FPs "work mainly in hospitals", which is factually incorrect, as FPs are trained to work especially in the district health services (first level of care which spans from community based care to first level hospital/district hospital care, and includes primary health care facilities such as community health centres and clinics).

7. The conclusions drawn do not ring true unfortunately. Although the authors state that their scoping review followed rigorous methodology, the analysis and interpretation does not match the findings of similar scoping reviews or observational studies included in this review. It is recommended that the authors review their findings and interpretations, and perhaps ask for another expert opinions from the African primary care context to corroborate their interpretations/theories. The authors are from two African countries only (Benin and DRC, both francophone), whereas the remaining authors are from Belgium. The concern is that this background of the authors may limit their perspective when engaging with the findings of the scoping review. Furthermore, it appears as if the authors are based in public health and do not have a primary care/family practice background? Please provide more information on the professional backgrounds of the authors. Three examples of incorrect/incomplete interpretation are presented here (there may be more issues to be identified by other reviewers):

7.1 For instance, stating that "GPs are left on their own", "FMLDs are poorly integrated, if at all, into local health systems and national health policies" (page 30, lines 492 - 493) appear to be a one-sided view and a simplistic assessment of a complicated issue spread across a diverse context. Such sweeping statements do not do justice to the intricacies faced by different country settings in Sub-Saharan Africa. It will be good to review the status of postgraduate FM training in Africa, as reassessed in 2019 during a Primafamed network meeting, which clearly highlights the various stages of change/engagement with this discipline across the network: https://phcfm.org/index.php/phcfm/article/view/2588/4132.

7.2 Also stating that "The GPs are concentrated in urban and wealthier areas, and the private GPs poorly contribute to health programs." (lines 456 - 457 on page 29) is untrue, as GPs also offer access in low/middle income settings (for example, South African "townships") where access to public sector primary care services is poor.

7.3 Statements on impact made by family physicians on page 22 (lines 370 - 372) are also incomplete. Whilst correlation analysis between FP supply and routine health indicators showed no impact due to low supply (reference 40), it should be noted that a cross-sectional study (reference 90) showed impact especially in district hospital regarding child healthcare indicators, as well as perceived impact from the perspectives of co-workers and district managers (references 86 and 73 respectively).

Reviewer #2: PONE-D-20-25494 Review (full review uploaded as an attachment)

General: This manuscript is the write up of a scoping review that was conducted to examine “…the current literature on First Line Medical Doctors (FLMD) in Sub-Saharan Africa (SSA)…” in order to “identify the knowledge gaps” (p.2 Protocol) about “what are the main characteristics and key issues of FLMD practices in SSA” (lines 92-93). Reviewing the literature over the past 19 years (2000-2019) in both English and French using a number of pre-identified search terms in five databases and purposively sampling some grey literature, the authors include 73 peer-reviewed (or original) research publications in their final analysis. The results indicate a range of publications, largely from South Africa (47%), and written in English (67%), although there are a number of papers (29%) from French-speaking Africa. There is great heterogeneity in the nomenclature of these “first line” doctors across SSA; and, the authors propose a bilingual classification based on how the literature refers to them. The authors subsequently construct three categories of FLMDs after examining their scope of practice and responsibilities whether in Anglophone Africa or in la Francophonie. Going systematically across four components of the health system (professional identity; governance; roles and activities; and, output and outcomes) for each type of FLMD (Family Physician; medicin generalist communautaire; or General Practitioner) the study reveals interesting observations about how, where, why and under what conditions these doctors work. These findings have important repercussions for national and regional health systems planners in Africa as well as newly qualified doctors deciding on career pathways. The paper also exposes significant gaps in research on medical doctors who function at primary care level, especially those in private practices across SSA and in rural areas as solo practitioners (as opposed to those placed there as part of government and/or NGO programmes). There is also little understanding of the return on investment in this cadre of health care worker vs the risks.

In summary, this article provides a refreshing look at task shifting in reverse and posits that when medical doctors move from referral centres into primary care spaces across Sub-Saharan Africa, there may be unforeseen consequences that could be managed with better role definition, governance and policies. It is critically important to understand the phenomenon of a shift of doctors away from referral hospitals into communities and clinical spaces while attending to undifferentiated patients at the primary care level, especially as these doctors engage patients and communities at points of first contact that were historically reserved for nurses. There are also potential policy decisions in resource constrained environments, since training and upskilling medical doctors to occupy roles in primary health care represents a costly investment. This is arguably a novel and relevant topic for a scoping review, and potentially can make substantive contributions towards addressing some of the human resources for health challenges, as well as the political and ethical ones.

Finally, this paper represents an ambitious project; and, the amount of work and reflection are obvious. The original research protocol is included as supplemental material, and indicates that this is part of a PhD from a candidate in Benin being supervised in Belgium. It is important for ISI and mainstream medical journals to publish articles about Africa, especially those conceived and written by Africans.

6. PLOS authors have the option to publish the peer review history of their article (what does this mean?). If published, this will include your full peer review and any attached files.

Reviewer #1: No

Reviewer #2: No

---

## [Author Response · Author response to Decision Letter 0]

10 Feb 2021

Responses to the reviewer 1

1. The term "FLMDs" sits uncomfortably - not sure if this may be due to translation? The accepted term is primary care doctors or primary care physicians (general/generalist doctors providing first-contact care at the primary level of care in the health system). The term family doctor is used by WONCA. This reviewer would strongly recommend that the authors consider changing to one of these terms, as "FLMDs" makes it difficult to relate to other research on the supply of primary care doctors or the primary care workforce (human resources for health). If the authors strongly feel that they would like to stick to the new term, they will have to make a stronger case, as the current text is unconvincing.

Our response 

We thank the reviewer for his suggestions. We did not use the terms “family doctors” or “family physicians” because they usually imply that the doctor would have a specific training in family medicine. The WONCA Europe indeed defines family doctors/physicians as “specialist physicians trained in the principles of family medicine/general practice (1)”. In Africa, the statement of consensus on family medicine stipulates that “a family physician has postgraduate training in Family Medicine” (2). Not all of the doctors described in this manuscript have received training in the principles of family medicine/general practice. 

The terms "primary care doctors" and "primary care physicians" refer to physicians who provide primary care, i.e. at the first level of contact between people and a professional health worker. The Institute of Medicine (IOM) defines primary care as "the provision of integrated, accessible health care services by clinicians who are accountable for addressing a large majority of personal health care needs, developing a sustained partnership with patients, and practicing in the context of family and community" (3). Our review is studying the practice of “medical doctors who work at the first line of healthcare delivery in SSA and provide all-around care to the population, without distinction based on the age, the sex or the clinical condition”. This definition is close to the definition of “primary care doctors” or “primary care physicians”. However, our review excludes doctors who work exclusively at the hospital level. And in some instances, primary care physicians do work only at the district hospital level (i.e. the second-line or referral level of healthcare) as some aspects of primary care can, rightly or wrongly, also be provided at this level. 

In conclusion, we propose using the term "primary care physicians" instead of "first-line medical doctors", as suggested. However, we made clear in the paper that we are limiting our analysis to "primary care doctors" working at the first-line of local healthcare delivery systems. 

Therefore, in the introduction section, we included a paragraph providing a clear definition of our study population (see line 105 to line 113 in the revised manuscript). 

2. On page 15, line 211, it is stated that the PRIMAFAMED network was created in 2017. This is factually incorrect. The origins of this network may be traced back to 1997. The name Primafamed was used for the first time in 2008. See editorial by founder Prof Jan de Maeseneer: https://phcfm.org/index.php/phc8fm/article/view/1603/2247.

Our response

Thank you for the correction. We corrected this. The current statement in the manuscript is: "The Primary Care and Family Medicine Education Network (PRIMAFAMED), created in 2008 and the collaborative projects that preceded it since 1997, helped in this harmonisation of FPs' training in SSA" (see line 280 to 282 in the revised manuscript).

3. On page 17, lines 250 onward, you might also wish to look at the national position paper drafted for the South African health department: https://scholar.sun.ac.za/handle/10019.1/99785; and a recent paper in the South African Health Review: https://www.hst.org.za/publications/South%20African%20Health%20Reviews/Chap%204%20Family%20Physicians.pdf. You will find that even though it appears that FPs are integrated in health and human resources policies, there is still a fair degree of ambiguity around the roles and scope of practice of FPs.

Our response

Thank you for this comment and the valuable documents provided. We agree that there is still ambiguity regarding the roles of FPs. Therefore, we have improved the analysis of the FPs' governance, and we revised the wording of this chapter to take this ambiguity into account. For example, we stated that in South Africa "Family physicians are acknowledged in several policy documents, although it appears that these documents still need to fully apprehend and integrate the FPs' roles" (see line 325 to 327). We also raised the fact that there is not always a clear understanding of the family physicians' roles (see line 333 in the revised manuscript), and that the family physicians may sometimes be inadequately positioned within the system (see line 335 to 336 in the revised manuscript).

4. On page 20, lines 312 - 314: whilst the Primafamed network has played a big role, it is incorrect to attribute all the expressions of family medicine in Africa to this network. There are many additional North-South and South-South influences over several decades, especially Nigeria, Uganda, Kenya and others.

Our response

Thank you for the comment. We corrected this part to highlight more these influences. See the following passage, from line 393 to 396 in the revised manuscript: "These roles were developed in the RSA, but other English-speaking African countries have adopted similar roles thanks to several North-South and South-South collaborations such as the PRIMAFAMED network or partnerships between African universities for implementing family medicine in several countries". We had also mentioned the North-South influences (especially the support from the Canadian and Belgian universities ) while describing the family physicians' professional identity (see line 198 to 201 in the original manuscript and line 267 to 270 in the revised manuscript). 

5. It has to be stated that the term general practitioner also carries a fair degree of ambiguity and confusion. In the international context, the terms general practitioners and family physicians/doctors are often used interchangeably for primary care doctors with 2 - 4 years of postgraduate training (see the Besrour series and the WONCA publications). Furthermore, in South Africa general practitioners refer to primary care doctors without postgraduate training who work in first level of care in the private healthcare sector, whereas medical officers refer to primary care doctors without postgraduate training who work across the levels of care in the public healthcare sector.

Our response: 

Indeed the term "general practitioner" has different meanings depending on the context. However, in the documents we reviewed, this term referred to primary care physicians who had neither a postgraduate training in a medical sub-speciality nor a postgraduate training in the principles of family medicine or primary care (fig 2 in the revised manuscript). Besides, in a number of countries (Benin, Senegal, Ethiopia for instance), "general practitioner" can refer to primary care physicians working in both the private and public sectors.

6. On page 27 in line 415, this is one example where the authors state that FPs "work mainly in hospitals", which is factually incorrect, as FPs are trained to work especially in the district health services (first level of care which spans from community based care to first level hospital/district hospital care, and includes primary health care facilities such as community health centres and clinics).

Our response

Thank you for raising this point. Some documents we reviewed indicate that by force of circumstances (such as the limited number of family physicians or the lack of clear understanding of their roles) and because of their high-quality training, a number of family physicians find themselves positioned at the district hospital level (whose central function is normally not to provide first contact care) or even at higher levels of care (4–6). However, we recognize that family physicians also work in community health centres and clinics (7, 8). That is why we have corrected the statement "FPs work mainly in hospitals" and similar statements by specifying that FPs work at all levels of the health care delivery system (see for example line 397 to 402 and line 514 in the revised manuscript).

7. The conclusions drawn do not ring true unfortunately. Although the authors state that their scoping review followed rigorous methodology, the analysis and interpretation does not match the findings of similar scoping reviews or observational studies included in this review. It is recommended that the authors review their findings and interpretations, and perhaps ask for another expert opinions from the African primary care context to corroborate their interpretations/theories. 

Our response

In this scoping review, we carefully analysed a wide range of peer-reviewed publications and grey literature. We mainly based our methodological approach on the methodology proposed by Arksey and O'Malley, which includes six steps: identification of the review question, identification of relevant studies, study selection, charting the data, collating, summarising and reporting the results and consultation of stakeholders. For the consultation of stakeholders, we presented the results to students in the Master in Public Health at the Institute of Tropical Medicine of Antwerp in September 2019. Many of these students were Africans and primary health care professionals, including primary care physicians. Also, we had informal discussions with other PhD students and African primary health care actors (from Uganda and Guinea in particular) to get further insights into primary care physicians' practices in their respective countries.

Moreover, although the data analysis was qualitative (making it difficult to rule out the authors' positionality completely), we endeavoured to ensure the validity and the reliability of the findings through an iterative analysis process. We adopted a transparent and systematic approach to synthesise the themes that emerged from the data analysis (see line 161 to 172 and line 232 to 260 in the revised manuscript), and we strived to accurately present in the results part the findings of the papers we reviewed. Also, although the review does not include opinion papers and conference proceedings, we used some of them (9–12) to discuss the findings, as a way of taking into account the point of view of other African primary care experts (see for example line 586 to 588 in the revised manuscript). 

We nevertheless recognize the importance of validating the findings of such an important and sub-Saharan Africa-wide review by a wide range of experts. Therefore we are grateful for your inputs and critical analysis. We also invited three additional colleagues from three different African countries to critically analyze the results and determine whether they resonate with their countries' reality. One of these colleagues (EW, from Kenya, who has conducted doctoral research on patient-centred care at primary care level in Uganda) is now included among the co-authors because of her substantial contribution to the manuscript's improvement. 

Finally, we recognize that we cannot draw definitive conclusions from this study because of the scoping review methodology's intrinsic nature. For this reason, we nuanced some of our conclusions because they are indeed hypotheses that need to be further tested by empirical data. For instance, instead of affirming that the primary care physicians are poorly integrated into local health systems, we have highlighted the existing ambiguity and pointed to the specific issues reported in the papers reviewed (see line 547 to 552 in the revised manuscript). Similarly, we emphasized more the need for further empirical research before concluding on the primary care physicians' contribution to sound primary care (see lines 516 to 517, 591 to 592 and 612 to 616 in the revised manuscript, for example).

7’. The authors are from two African countries only (Benin and DRC, both francophone), whereas the remaining authors are from Belgium. The concern is that this background of the authors may limit their perspective when engaging with the findings of the scoping review. Furthermore, it appears as if the authors are based in public health and do not have a primary care/family practice background? Please provide more information on the professional backgrounds of the authors. 

Our response

Most of the authors (KB, SB, EW, JPD, JDLP, LA and BC) have worked (or even currently work) at primary care level. The authors from Belgium have a huge experience in low-and-middle-income countries, especially in sub-Saharan Africa, and have an accumulated experience of many decades on primary health care in Europe, Africa and Asia. 

The first author (KB) is a public health doctor indeed, but she is also a General Practitioner and worked for ten years (2008 to 2018) as a primary care physician at the first-line of healthcare delivery in Benin. Even if she had to temporarily suspend her clinical activities because of her PhD program, she is still in close touch with the primary care physicians in the field. Her PhD research focuses precisely on the analysis and improvement of the practices of these primary care physicians. 

JDLP is professor of family medicine at the University KU Leuven in Belgium. He is a family physician for decades, and he is the leader of a writing group on mental health in Primary Care within the European Forum for Primary Care (EFPC). In his unit, several PhD programs are conducted in low-and-middle-income countries.

JK and SB are Congolese physicians with a very good knowledge of the local health systems' organisation in their country. SB is a district medical officer, and he has good insights into the practices of primary care physicians in his health district. Moreover, he recently conducted a study on primary care physicians in Kisangani, DR Congo. This study was recently published in French in the African Journal of Primary Health Care and Family Medicine. 

JPD is a public health doctor and health policy and systems researcher from Benin. He has a good command of several research methodologies, including the scoping review methodology. He has also worked as a primary care physician at the beginning of his medical career. 

EW is a registered nurse from Kenya and a health systems researcher and quality of care change agent. She is currently writing up and finalizing her PhD thesis on understanding the challenges and opportunities in implementing patient-centred primary health care in Uganda. She has a good experience of the health care organization within health districts in Kenya and Uganda. 

LA is a clinical doctor and a post-doctoral researcher at the Institute of Tropical Medicine in Antwerp, Belgium. He is from Belgium and is a trained Family Physician who worked for several years in a primary care practice. Later he worked for more than ten years in Africa, mainly in Zimbabwe but also in Kenya, DR Congo and Morocco. In Zimbabwe, he held several positions at primary health care level, including medical officer and district medical officer. 

MZ is a professor of internal medicine at the University of Abomey Calavi, in Benin. He has been teaching to undergraduate medical students for more than twenty years. He is also involved in the training of other primary care clinicians, and he has been the deputy director of Benin's national nursing school for three years. Besides his teaching and clinical activities, he contributes to several initiatives for primary care improvement in Benin. 

BC is a professor of public health at the Institute of Tropical Medicine Antwerp. He has over 30 years of experience in the organisation and management of local health systems and primary health care (including family medicine), in Belgium and a wide range of African and Asian countries. He is currently coordinating health systems strengthening projects in the Democratic Republic of Congo, Guinea (Conakry), India, Mauritania and Uganda.

Three examples of incorrect/incomplete interpretation are presented here (there may be more issues to be identified by other reviewers):

7.1 For instance, stating that "GPs are left on their own", "FMLDs are poorly integrated, if at all, into local health systems and national health policies" (page 30, lines 492 - 493) appear to be a one-sided view and a simplistic assessment of a complicated issue spread across a diverse context. Such sweeping statements do not do justice to the intricacies faced by different country settings in Sub-Saharan Africa. It will be good to review the status of postgraduate FM training in Africa, as reassessed in 2019 during a Primafamed network meeting, which clearly highlights the various stages of change/engagement with this discipline across the network: https://phcfm.org/index.php/phcfm/article/view/2588/4132.

Our response

We tried to bring out this complexity by presenting and discussing the variability in primary care physicians' practices in Sub-Saharan Africa. The first part of our discussion concerns this variability and complexity (line 503 to 545 in the revised manuscript). Moreover, throughout the manuscript, we highlighted as much as possible the differences between countries, and we exposed the differences between the different types of primary care physicians in Sub-Saharan Africa. For example, we exposed that even though several papers reported that GPs in the private sector are business-oriented, other papers indicate that these GPs can opt for a publicly oriented service delivery (line 317 to 319 in the revised manuscript). 

However, as explained above, we improved the manuscript by avoiding too general conclusions.

7.2 Also stating that "The GPs tend to be concentrated in urban and wealthier areas, whereas private GPs do not always contribute to public health programs." (lines 456 - 457 on page 29) is untrue, as GPs also offer access in low/middle income settings (for example, South African "townships") where access to public sector primary care services is poor.

Our response

Thank you for raising this point. We revised the wording of this part. We now stated that: "there were also concerns regarding the accessibility, the comprehensiveness and the continuity of the services offered by the GPs" (see line 588 to 590 in the revised manuscript). Indeed, several studies have reported that private GP services are mainly accessible to people who can afford it (see line 486 to 487 in the revised manuscript). This may be because many private GPs need to be paid out-of-pocket or via private health insurance, which some people cannot afford. Other studies also reported a focus on curative care among some GPs (see line 490 to 491). 

We nevertheless recognize that these issues are not necessarily found among all the GPs. And we specified it in the revised manuscript (see line 491 to 493 in the revised manuscript). Besides, we have highlighted that the private GPs can offer an alternative when the public primary care services are poorly available (see line 392 to 395 in the original manuscript and lines 484 to 486 in the revised manuscript). 

7.3 Statements on impact made by family physicians on page 22 (lines 370 - 372) are also incomplete. Whilst correlation analysis between FP supply and routine health indicators showed no impact due to low supply (reference 40), it should be noted that a cross-sectional study (reference 90) showed impact especially in district hospital regarding child healthcare indicators, as well as perceived impact from the perspectives of co-workers and district managers (references 86 and 73 respectively).

Our response

We presented the perceived impact of family physicians and their positive effect on the access to care, the clinical processes and some health outcomes, especially those related to mother and child health (line 411 to 416 in the original manuscript).

However, we recognize that the statement "The rare studies that have evaluated the impact of FPs on the performance of local health systems were performed in the RSA, and they could not demonstrate any impact" (line 420-422 of the original manuscript) was not precise enough. We were specifically referring to the family physicians' impact on health outcomes such as child mortality, maternal mortality, mother to child HIV transmission rate, etc. We thus revised this part as follows: "We found only a few observational studies that have evaluated the impact of FPs on health outcomes (such as child mortality, maternal mortality or mother to child HIV transmission rate). These studies were limited to the Republic of South Africa. They could not demonstrate a significant impact of the family physicians on health outcomes, despite the positive impacts reported on clinical processes and accessibility. This was attributed to the limited number of FPs" (see line 457 to 461 in the revised manuscript).

We have also reorganised the outputs/outcomes section to separately present the primary care physicians' impact on the quality of primary care and their impact on health outcomes (from line 444 to 494 in the revised manuscript). For instance, for the specific case of the family physicians, we first presented the impact of the family physicians on the access to care, the quality of the clinical processes, and other aspects of primary care (line 445 to 457 in the revised manuscript). We then discussed their impact on health outcomes (see line 457 to 461 in the revised manuscript). 

References

1. WONCA Europe. The European definition of general practice / family medicine- 2011 Edition [Internet]. Europe; 2011. Available from: http://www.woncaeurope.org/.

2. Mash R, Reid S. Statement of consensus on Family Medicine in Africa. Afr J Prim Heal Care Fam Med. 2010;2(1):4–7. DOI: 10.4102/phcfm.v2i1.151.

3. Committee on the future of primary care. Primary Care: America’s Health in a New Era. Donaldson MS, Yordy KD, Lohr KN, Vanselow NA, editors. National Academy Press. Washington D.C.; 1996. https://www.ncbi.nlm.nih.gov/books/NBK232643/.

4. Flinkenflögel M, Sethlare V, Cubaka VK, Makasa M, Guyse A, De Maeseneer J. A scoping review on family medicine in sub-Saharan Africa: Practice, positioning and impact in African health care systems. Hum Resour Health. 2020;18(1):1-18. DOI: 10.1186/s12960-020-0455-4.

5. Besigye IK, Onyango J, Ndoboli F, Hunt V, Haq C, Namatovu J. Roles and challenges of family physicians in Uganda: a qualitative study. Afr J Prim Health Care Fam Med. 2019;11(1):1-9. DOI: 10.4102/PHCFM.V11I1.2009.

6. Mash R, Von Pressentin KB. Strengthening the district health system through family physicians. South African Heal Rev 2018; 2018: 33–39. DOI: 10.10520/EJC-1449142b2d.

7. Von Pressentin KB, Mash RJ, Esterhuizen TM. Examining the influence of family physician supply on district health system performance in South Africa: an ecological analysis of key health indicators. Afr J Prim Health Care Fam Med. 2017 Apr;9(1):1-10. DOI: 10.4102/phcfm.v9i1.1298. 

8. Von Pressentin KB, Mash RJ, Baldwin-Ragaven L, Botha RPG, Govender I, Steinberg WJ , et al. The Influence of family physicians within the South African district health system: a cross-sectional study. Ann Fam Med. 2018 Jan;16(1):28-36. DOI: 10.1370/afm.2133.

9. Reid S, Mash B, Thigiti J, Nkombua L, Bossyns P, Downing R, et al. Names and roles for the generalist doctor in Africa. Afr J Prim Health Care Fam Med. 2010;2(1):1-5. DOI: 10.4102/phcfm.v2i1.242.

10. De Maeseneer J. Scaling up Family Medicine and Primary Health Care in Africa : Statement of the Primafamed network , Victoria. 2013;5(1):1-3. DOI: 10.4102/phcfm.v5i1.507. 

11. De Maeseneer J, Flinkenflogel M. Primary health care in Africa: do family physicians fit in? Br J Gen Pract. 2010 Apr;60(573):286-92. DOI: https://doi.org/10.3399/bjgp10X483977.

12. Moosa S, Peersman W, Derese A, Kidd M, Pettigrew LM, Howe A, et al. Emerging role of family medicine in South Africa. BMJ Glob Health. 2018;3:2-4. DOI: 10.1136/bmjgh-2018-000736.

 

Responses to reviewer 2

1 PONE-D-20-25494 Review General: This manuscript is the write up of a scoping review that was conducted to examine “…the current literature on First Line Medical Doctors (FLMD) in Sub-Saharan Africa (SSA)…” in order to “identify the knowledge gaps” (p.2 Protocol) about “what are the main characteristics and key issues of FLMD practices in SSA” (lines 92-93). Reviewing the literature over the past 19 years (2000-2019) in both English and French using a number of pre-identified search terms in five databases and purposively sampling some grey literature, the authors include 73 peer-reviewed (or original) research publications in their final analysis. The results indicate a range of publications, largely from South Africa (47%), and written in English (67%), although there are a number of papers (29%) from French-speaking Africa. There is great heterogeneity in the nomenclature of these “first line” doctors across SSA; and, the authors propose a bilingual classification based on how the literature refers to them. The authors subsequently construct three categories of FLMDs after examining their scope of practice and responsibilities whether in Anglophone Africa or in la Francophonie. Going systematically across four components of the health system (professional identity; governance; roles and activities; and, output and outcomes) for each type of FLMD (Family Physician; medicin generalist communautaire; or General Practitioner) the study reveals interesting observations about how, where, why and under what conditions these doctors work. These findings have important repercussions for national and regional health systems planners in Africa as well as newly qualified doctors deciding on career pathways. The paper also exposes significant gaps in research on medical doctors who function at primary care level, especially those in private practices across SSA and in rural areas as solo practitioners (as opposed to those placed there as part of government and/or NGO programmes). There is also little understanding of the return on investment in this cadre of health care worker vs the risks. In summary, this article provides a refreshing look at task shifting in reverse and posits that when medical doctors move from referral centres into primary care spaces across Sub-Saharan Africa, there may be unforeseen consequences that could be managed with better role definition, governance and policies. It is critically important to understand the phenomenon of a shift of doctors away from referral hospitals into communities and clinical spaces while attending to undifferentiated patients at the primary care level, especially as these doctors engage patients and communities at points of first contact that were historically reserved for nurses. There are also potential policy decisions in resource constrained environments, since training and upskilling medical doctors to occupy roles in primary health care represents a costly investment. This is arguably a novel and relevant topic for a scoping review, and potentially can make substantive contributions towards addressing some of the human resources for health challenges, as well as the political and ethical ones. Finally, this paper represents an ambitious project; and, the amount of work and reflection are obvious. The original research protocol is included as supplemental material, and indicates that this is part of a PhD from a candidate in Benin being supervised in Belgium. It is important for ISI and mainstream medical journals to publish articles about Africa, especially those conceived and written by Africans. 

Our response

Thank you for reviewing this article and for the positive appreciation of our work. Your further comments significantly contributed to improving our work. We are grateful to you for highlighting that the progressive shift of medical practice from mainly referral centres to primary care spaces in Sub-Sahara Africa may have unforeseen consequences if not properly managed. We hope that our findings will shed light on this issue and trigger further research and actions to guide primary care physicians' practices towards better primary care in SSA. 

Major: There are several significant conceptual framings to the manuscript and methodological decisions that, to my mind, require clearer explanation and more rigorous justification in the text. I have outlined these in the first three points below, followed by other major issues in no particular rank order of importance: 

1. Including a more robust definition of the "first-line medical doctor" (FLMD) in the introduction is critical. Although FLMDs are the cornerstone of this study, this is not a common term in the medical literature, nor is it one with which most readers (even Englishspeaking medical generalists) would be familiar. A clear definition of FLMDs will enable the article to reach the broader medical community and articulate with conversations in other journals about the deployment of primary care doctors in SSA and their scope of practice. To illustrate the limitation of the use of FLMD: when I tried to search within the PLOS ONE option for similar publications on MEDLINE, this is the message I received: "Your search was processed without automatic term mapping because it retrieved zero results." Similarly, when I used "first line medical doctor" in PUBMED on 2020/10/25, this is the response I got: "Quoted phrase not found." In Google scholar (albeit not ideal), there were eight results (five unique, three duplicates, one from Quebec, the rest European). For a predominantly English audience, this term needs more unpacking, and will probably necessitate revision of the title of the manuscript to in order to resonate with a wider (and more appropriate) audience.

Our response

We acknowledge the limitations of using the term "first-line medical doctor". We decided to replace it with "primary care physicians" (PCPs) to better relate with other research on the primary care workforce. We also included in the introduction section a paragraph providing a clear definition of the study population (see line 105 to 113 in the revised manuscript). We formulated this paragraph as follows: 

"In our review question, PCPs are defined as medical doctors who work at the first line of healthcare delivery in SSA and provide all-around care to the population, without distinction based on the age, the sex or the clinical condition. We excluded doctors who work exclusively at the hospital level because the novelty of the phenomenon we are studying lies on the shift of medical practice from hospitals to the first-line. We also limited the study to physicians who provide all-round care to the population because the key function of the first-line is to provide primary care, which is defined by the Institute of Medicine as "the provision of integrated, accessible health care services by clinicians who are accountable for addressing a large majority of personal health care needs, developing a sustained partnership with patients, and practising in the context of family and community". 

We also defined the first-line, in relation to health districts and primary health care (see line 60 to 66): 

“In Sub-Saharan Africa (SSA), many countries operationalise PHC within health districts, which encompass a network of formal health facilities, community-based services, and other supporting services and health programs. The formal healthcare delivery platform includes small to medium size public and private facilities (the first line) which should normally be the first entry point in this platform and should deliver primary care, dealing with the majority of the population's health needs. These first-line facilities (called dispensaries, health centres, community health centres or clinics, depending on the context) are supported by a district hospital (the second line) which is the first referral level.”

Finally, we revised the title based on all of the above. The current title we propose is: "The expanding movement of primary care physicians operating at the first line of healthcare delivery systems in sub-Saharan Africa: A scoping review". 

 2. There is an uneasy linkage between FLMDs in the manuscript (whose role is largely clinical) and the pillars of health systems research. I could not find sufficient justification for the use of the health systems dynamics framework (Reference 27) as the sole basis for categorising the articles and creating a data extraction form (lines 117 – 120). Since the data extraction form does not appear to be included in either the body of the paper or in the supplemental files, it is unclear exactly what data were extracted from which articles onto an Excel spreadsheet. This omission also speaks to the limited availability of data, which is something PLOS ONE requires. Furthermore, it seems that this framework gets jettisoned later in the Discussion and Conclusion, failing to take full advantage of the model in its ability “to deal with complexity/its dynamic character/and the values embedded in it.” There is no convincing explanation as to why the framework was employed to begin with, since “health systems” is not included in the key words, nor were these different components (“governance”; “resources”; “service delivery”; “relationships with the community context”; “values on which the health system is based”; and, “outcomes”) used in the search strategy at the outset. Rather than allowing the themes of the scoping review to emerge organically and synergistically through iterative discussions among the authors (as it seems might have been the case anyway in Table 5), the method of using the ‘health systems dynamics’ framework could have restricted the reviewers’ vision of appreciating the actual subject matter being covered by these 73 articles. Without access to the Excel data extraction form, and only the S3 Table (List of papers included), it is difficult to understand the data items and the data charting process. As a reviewer, it would have been important to confirm the fields on the data extraction form, and look for alignment on several of the articles through a trial run. At the moment, none of this is replicable. Being familiar with some of the literature, it seems that several important (and statistically significant) outcomes were not adequately captured. 

Our response

In the revised manuscript, we explained why we used the health systems dynamics framework to construct the data extraction form (see line 150 to 156). 

Although the role of PCPs who work at the first-line is mainly clinical, their practice should contribute to improving the performance of the whole health system. Moreover, the other components of the health system (for instance the governance arrangements, the general context or the resources available) influence the roles assigned to these PCPs, their activities and even the results achieved. For example, we could see that the family physicians' efforts to apply primary care values are often undermined by the hierarchical culture in their context or by a high workload (which is sometimes due to the fact that they endorse the role of other specialists who are lacking in district hospitals). So, using the health systems dynamics framework as a starting point for constructing the data extraction form helped us systematically look at the information on each of the health system elements in relation to the practice of PCPs.

Furthermore, the framework guided our analysis of the data extracted from the documents reviewed. Some of the themes and dimensions used in this review for analysing the PCPs' practices (table 4 in the revised manuscript) are elements of the health system dynamics framework or derived from them (for example governance, values and outcomes). The additional themes and dimensions emerged from the data in two ways. First, apart from the predefined fields in the data extraction form, the reviewers extracted additional data when deemed necessary. Second, during the data analysis and synthesis process, the themes were iteratively discussed and refined, allowing for identifying sub-themes (for instance, the historical pathway) and grouping some themes into the dimensions presented in table 4. We have better explained this analysis process in the methodology section (see line 162 to 172 in the revised manuscript).

Finally, although we did not directly use the health system dynamics framework in the discussion and conclusion parts, we strived to bring out the relationships between the various elements, as recommended for an analysis taking a health system perspective. This helped us link the governance of PCPs to their activities and to their contribution to primary care, in the discussion section. For example, we highlighted the potentials links between the policy guidance, the activities of the PCPs and the outcomes of their practices (see line 560 to 566 in the revised manuscript). We did a similar analysis in the original manuscript from line 430 to 435. 

3 . Aside from not using any of the key components of the ‘health systems dynamics’ framework above in the search strategy, such as “service delivery”, there seem to be other aspects of the search strategy that are not clear or do not align with the stated objectives of the scoping review extracted below from the Protocol (S1 Text): - This scoping review aims at […] providing an overview of the current literature on First Line Medical Doctors (FLMD) in Sub-Saharan Africa and identifying the knowledge gaps. (p 2 Protocol) - General objective: the general objective is to map the existing knowledge on First Line Medical Doctors (FLMD) in Sub-Saharan Africa and to identify the key issues in this area. - Specific objectives and research questions: a. Determine the various types of medical Doctors operating at the level of first-line in SSA b. Map the key dimensions that have been studied on FLMD in Sub-Saharan Africa (training, resources, services delivery, leadership and government [sic] arrangements, etc.) c. Determine the key issues related to FLMDs’ practice in Sub-Saharan Africa My specific questions in this regard are: 

3.1 While the justification for starting the search from the year 2000 is based on the prevalence of articles dating from that year (line 102), there may be foundational publications that could have been missed. Having been part of early debates that dealt with questions of professional identity and the need for a values-driven medical practice at primary care level in SSA ("one family"/"one doctor"), it seems prudent to go back to the 1990's, or provide a stronger justification for limiting the search to publications dating from 2000 other than convenience. Even if the year 2000 is retained, this decision could be dealt with in the discussion, with reference to earlier seminal articles not included in the scoping review, in order to apply a historical/ contextual dimension and evolutionary analysis to what has been published over this nearly 20 year period and how it has developed over time.

Our response

Apart from the prevalence of articles dating from 2000, another reason for starting the search in 2000 is the fact that some of the papers written after 2000 provided information on the development of the primary care physicians in sub-Saharan Africa going back to earlier years. For instance, the papers by Dugas et al. “La construction de la médecine de famille dans les pays en développement” (1) and by Desplats et al. "Pour une médecine générale communautaire en première ligne" (2) provide information and insight on the development of the "médecins généralistes communautaires". Concerning the family physicians, the papers by Hellenberg et al. "Family medicine in South Africa: where are we now and where do we want to be?"(3), Moosa et al ."The views of key leaders in South Africa on implementation of family medicine: critical role in the district health system" (4), and other papers provided a summary of the development of family medicine in the Republic of South Africa. We also found papers describing the development of family medicine in other countries such as the paper by Makwero et al “A. Family medicine training and practice in Malawi: history, progress, and the anticipated role of the family physician in the Malawian health system” (5). 

We nevertheless acknowledge the relevance of applying a historical and contextual dimension to the analysis to fully understand the practices of PCPs in Sub-Saharan Africa, especially how the specific African context and the early discussions have shaped these practices and the values that should drive them. Therefore, we discussed the role that the historical pathway of each of the three categories of PCPs may have played in the current expression of their practices (see line 520 to 533 in the revised manuscript). Furthermore, we had already discussed how the expectations that African stakeholders have of physicians might have influenced their activities and performance regarding primary care (see line 593 to 600 in the revised manuscript).

 3.2 Why was there a decision to leave out government policy documents on the deployment of FLMDs? It seems that inclusion of this type of grey literature would be important in order to answer the research question/objectives of the study. Through government documents or discussion papers, each country in SSA might have been represented in ways that publishing in peer-reviewed indexed journals with high APCs would never be possible. The authors might perhaps consider explaining this exclusion and adding it to the section on limitations. 

Our response

We only had access to government policy documents for some countries, and it was not easy to get them for the other countries without a primary data collection. So, we decided to leave out these documents in order to avoid a potential selection bias. However, we definitely agree that the exclusion of government policy documents is a limitation of this study. We acknowledged it in the strengths and limitations section, from line 626 to 627 in the revised manuscript. 

3.3 Also on the matter of grey literature, it is unclear as to why other NGOs (besides Sante Sud) or University Medical Faculties or professional groupings of primary care practitioners (besides PRIMAFAMED) or religious groups working in SSA were not purposively sought out. As notional examples, I am thinking of government-to government agreements between Cuba and countries in SSA to supply first line doctors; MSF and other humanitarian relief agencies; CHAI support to the training of medical doctors in Rwanda; etc. 

Our response

We acknowledge that we did not comprehensively search for the grey literature, and we explicitly include this in the "limitations" section of the paper (see line 628 to 630 in the revised manuscript). However, throughout the review process, we attempted to mitigate the impact of this limitation through a careful review of the papers, including the introduction part, and a careful screening of their reference lists. We also mentioned in the revised manuscript, the efforts from the African governments to supply first-line doctors, including the government-to-government agreements between Cuba and countries (see line 315 in the results section of the revised manuscript)

3.4 Since the colonisation of SSA was not restricted to the English and the French, why was the search not trilingual to include Portuguese, or even other languages such as German, Dutch and Flemish? There would be many SSA countries with such colonial histories, and legacies of these relationships, whose publications might have been missed. Again, perhaps something to include in limitations? 

Our response

Thank you for the pertinent comment. We have addressed this in the limitations section on lines 634 to 636 in the revised manuscript. 

3.5 I am curious as to why an expanded search was not undertaken once certain framings came to light through the analysis of the articles. For example, after the French terms medecin de campagne or generaliste communautaire came up, why were the English translations/interpretations not subsequently incorporated into the search strategy? “Rural doctor” yields a great deal of results that do not appear to be included in the 73 articles. Likewise, the literature on “community service doctors” in the South African context, and perhaps others, is missing, although it appears similar to the generaliste communautaire. 

Our response

Thank you for the useful advice. We performed an additional search on MEDLINE on January 15, 2021, by including the keywords "community doctors" and "rural doctors" in the search strategy. This yielded 53 results, of which some were duplicate of papers already reviewed. Three additional papers met the inclusion criteria and were included in the review. One of the papers was about the physicians trained in Cuba. We also found some papers on the community services doctors, but, unfortunately, they did not meet the inclusion criteria (mainly because they were reported to work in district hospitals). 

We also performed an update search with the previous search strategy, starting from June 2020. Two additional papers met the inclusion criteria. 

3.6 As well, in the search strategy, it is not clear why a manual review of several journals critical to this subject matter was not undertaken. For example, purposive sampling of Rural and Remote Health, African Journal of Primary Care and Family Medicine and Human Resources for Health might have yielded results that were more focussed and/or relevant to the key questions of the scoping review. 

Our response

We did a manual search of the following journals: Rural and Remote Health, African Journal of Primary Care and Family Medicine and Human Resources for Health in January 2021. We also found several duplicates there. However, we got three additional papers. Thank you for suggesting this manual review. 

All the additional searches performed provided eight papers that were used in the revised manuscript to support the previous finding or refine the analysis. However, they did not introduce significant changes in the findings.

3.7 Finally, the correct reference for the PRISMA extension for Scoping Reviews (reference 26) should be Tricco AC, Lillie E, Zarin W, O'Brien KK, Colquhoun H, Levac D, et al. PRISMA Extension for Scoping Reviews (PRISMAScR): Checklist and Explanation. Ann Intern Med; 169:467–473. doi: 10.7326/M18-0850. Although the correct PRISMA-ScR document is included as “S2 Text”, it is still missing the appropriate citation in the body of the manuscript (JBI has taken down the link in #26) as well as permission to re-print from St Michael’s. Using Tricco et al as a guide will assist in revising the above points as well as addressing some of what follows below

Our response

Thank you for the correction. We provided the appropriate citation for the PRISMA extension for Scoping Reviews (Tricco and al, 2018, see reference N° 32).

 Also, during the conception phase of this study, we used several methodological guides (see reference list of the protocol, S1-text) including: "Tricco AC, Soobiah C, Antony J, Cogo E, Macdonald H, Lillie E, et al. A scoping review identifies multiple emerging knowledge synthesis methods, but few studies operationalize the method. J Clin Epidemiol. 2016;73:19–28" and "Alliance for Health Policy and Systems Research. Evidence Synthesis for Health Policy and Systems : a Methods Guide. Langlois E V, Daniels K, Akl EA, editors. Geneva: World Health Organization; 2018". We included them in the revised manuscript references (see lines 118 to 120 in the revised manuscript and references N° 29 and N° 30). 

 4. The Flow Chart, Figure 1, is missing critical information that makes it difficult for the reader to follow why some articles were excluded at certain points and why others were retained. In addition, aside from a brief mention of the review process in the lines 131-132, a more robust narrative should be included in the Results section to describe the flow chart. 

Our response

We included a better narrative of the study selection process in the Results section, from line 182 to 186 in the revised manuscript. 

Specifically: 4.1 The disaggregated sources of articles should be included in the “Identification” step, with the exact numbers sourced from each of the five databases, rather than a combined total of n=3939. “Other sources” should also be itemised at this point. 

Our response

Thank you for the suggestion. We provided the disaggregated sources of articles, and we itemised the "other sources" at this point (see figure 1). 

4.2 The reasons for excluding 3844 records through screening the title and abstract are not clearly articulated in either the text or in the flow diagram. Lines 112-113 indicate, that “after removing the duplicates, two of the authors independently assessed the titles, abstracts, and full-text of the articles with predefined selection criteria (Table 2)”, thereby appearing to merge the steps of full-text review (“Eligibility”) with that of “Screening”. This distinction needs to be maintained, rather than simply stating that 3844 records were excluded without providing a clear rationale or criteria for elimination. 

Our response

These issues have been addressed in the revised manuscript and the flow chart. We provided more details on the review process in the methods section from line 139 to 145 in the revised manuscript. We also included in the flow chart the reasons for excluding the 3844 records (see figure 1 in the revised manuscript). 

4.3 The 70 articles that were excluded based on the eligibility criteria might be valid; however, the restrictions are not completely aligned with what is outlined in Table 2. For example, one of the exclusion criteria is “papers published before 2000” whereas in the flow 5 diagram, manuscripts were excluded because “data [were] collected before 2000”. The reasons (inclusion/exclusion criteria) need to be consistent and strictly adhered to throughout this process.

Our response

We actually integrated the papers which data were collected before 2000. One of the reviewers excluded them, but after discussion, they were included. We forgot to remove this reason from the flow chart. We apologize for this error which we have corrected in the flow chart.

 As stated above, opinion papers, commentaries and conference or workshop reports might have been valuable to retain, given that this is a scoping (rather than a systematic) review. The focus of the scoping review on “peer reviewed” original research requires further justification, as indicated in Point 3 above.

Our response

We excluded opinion papers, commentaries and conference or workshop reports because they usually express the writer's or the participants' opinion, rather than reporting the actual practice. However, we used several of these papers in the introduction and the discussion sections. For example, we used some commentaries to discuss the primary care physicians' impact on the primary care performance (see line 587 to 588 and the references N° 118 and N° 127 in the revised manuscript). 

 Furthermore, we did not limit the scoping review to peer-reviewed papers. We included several pieces of grey literature, especially those reporting country cases or other case studies. 

 Another exclusion criterion from Table 2 “Papers relating to MDs working exclusively in hospitals” might have inadvertently eliminated research about important cadres of first line doctors, especially those working in rural areas in SSA, where there may only be a district or mission hospital serving a sparsely populated area through outreach initiatives rather than a formalised network of other PHC facilities (such as clinics or health posts). 

Our response

This is a relevant point, indeed. However, although we did our best to include the hospital doctors who have an additional community practice, it was not easy to distinguish all the nuances in the practices of the PCPs based on the papers. We thus included this point in the limitations section (see line 630 to 633 in the revised manuscript). 

4.4 In the "Inclusion" step, the addition of 17 articles through citation tracking and six articles through a new search should be indicated with arrows below the box of "Full-text articles eligible" (n=50), as well the exact date of the new search (not only the year). These should then be merged, to add 23 articles to the 50, for the total of 73.

Our response

We corrected this on the flow chart. Thank you for the suggestion. 

 4.5 Finally, although additional tables and diagrams are used in the manuscript to describe the types of articles included, it might be useful to add a basic characterisation of the types of sources that made the cut. This could go in below the “Papers included in the scoping review”, and might be a classification of the number of those written in French vs English, or about specific countries in SSA (therefore removing Figure 2), or by type of methodology (quantitative, qualitative and mixed methods)—to better (and more graphically) indicate the heterogeneity and variety of the final selections. 

Our response 

In the results section, we included on page 12 a table (table 2), which provides a characterisation of the papers included. This table presents the number of papers written in French vs English, a classification of the papers by methodology and the number of papers about each sub-Saharan African region. 

5. While the article requires copy editing to address grammatical, spelling and punctuation errors, there are important clarifications of terminology that might have gotten lost in translation from French to English. Most striking amongst these are: 

5.1 In the Results section, the use of “denomination” to indicate “nomenclature” or “taxonomy” to describe the different types of FLMDs discussed in the literature. This should be addressed in lines 138 – 140; 142, and in Table 3. 

Our response

Thank you for the suggestion. We replaced the term “denomination” by "nomenclature" or "name" "or designation" as appropriate (see for instance line 197, line 201, line 202 and table 3 in the revised manuscript).

5.2 Furthermore, some aspects of categorising these doctors into groupings based on the descriptions of their activities seem problematic (Figure 3). For me, combining medical officers (MOs) and GPs misses the point of general practitioners’ largely private practices being run as small businesses with pharmacies and dispensing options as well as niche or boutique foci, such as aesthetic medicine: they are inherently unregulated. Ironically, however, at least in the South African context, GP practices are stable caring for families and communities across generations. Medical officers operating at first line, on the other hand, are never self-employed, and they often work publically. Their contractual relationships are either with either government or another regulatory body, such as a health insurer or syndicate of health care providers. Like the generalistes communautaires (and community service doctors), MOs are usually transitory remaining in a post until an opportunity for specialisation opens up. The groupings, and conclusions subsequently drawn, do not seem plausible. Therefore, Figure 3 needs revision. 

Our response

Based on the papers reviewed, we did not find sufficient arguments for making medical officers a separate category. Of the 7 papers relating to medical officers, only 2 correctly described their practices. The others just evoke certain aspects of this practice. So, if we had made the medical officers a separate category, it would have been difficult to characterize them properly. Moreover, the practices of PCPs at the first-line in the public sector may vary, depending on the context. For instance, in Benin, they appear to be more stable than the private GPs (unlike in South Africa). Thus, further research would be needed to define the differences between public and private GPs more accurately. We nevertheless acknowledge that describing the GPs category as one homogenous lot would be misleading. Therefore, we tried, whenever possible, to bring out the possible differences between public and private GP. Examples in the original manuscript can be found at lines 240 to 242, 245 to 246, 289 to 291 and 342 to 344. Examples in the revised manuscript can be found at lines 311 to 315, 316 to 319, 367 to 369 and 426 to 429. 

Additionally, we highlighted in the discussion section these possible differences between the public and the private GPs and the need for more research to verify and specify these differences (see line 516 to 517 in the revised manuscript). 

6 6. Discussion: The phrase "a multiform phenomenon" (line 403) needs rewording. Care must also be taken in terms of sweeping generalisations "…in the RSA, FPs are trained for four years and work mainly in hospitals…" (lines 415 – 416). There are regional differences within RSA, with provinces deploying FPs to community health centres and/or sub-districts rather than to district hospitals. Similarly, lines 425 – 429 starting with "The FLMDs are poorly integrated into local health systems and national health policies […]" might be over-stated from the articles available. Simply because the authors "did not find a clear and overarching governance framework to guide FLMDs" does not mean that in certain contexts it does not exist. It simply means that the authors did not find it! This brings me to the point of the purposes and limitations of this type of study and the conclusions that can in fact be drawn from a scoping review. There is no critique in this manuscript about what types of research gets published and indexed from the Global South, and especially from SSA. In this review, South Africa/PRIMAFAMED/NGO Sante Sud dominate and therefore, simply put, get to tell the story. It would behove the authors to highlight the research gaps more clearly and raise issues around publication bias. Certain journals have annual quotas for articles from Africa— and perhaps other parts of the Global South. APCs are prohibitive and most journal publications are in English. Therefore, the current conclusion over-reaches in places about what can be said from this scoping review, and should be re-framed in the tone highlighted in line 459, "…more empirical evidence is needed before drawing final conclusions." The study can only comment about gaps in the literature that describe what is, or perhaps more importantly, what is not there. 

Our response

Thank you for these useful comments. We agree with them, and we have carefully revised the tone of the whole article, especially the findings and the discussion sessions. We paid particular attention to avoiding sweeping generalization and highlighting the research gaps. For example, we presented the situation of the family physicians in South Africa more accurately. We specified that they could work at all levels of the healthcare system (see lines 397 to 401 and 513 to 514 in the revised manuscript). We also discussed the issues related to the primary care physicians' governance and their contribution to primary care performance, without drawing overarching conclusions. For instance, from line 547 to 552, we summarised the governance issues pointed out by the paper reviewed without concluding that there is a "poor governance". 

Furthermore, we highlighted the research gaps more clearly (see lines 537 to 545, 612 to 616 and 648 to 651 in the revised manuscript), and pointed to the potential role of publication bias in explaining these research gaps ( line 540 to 545 in the revised manuscript). 

Finally, we removed the phrase "a multiform phenomenon" and replace it with "heterogeneity in the PCPs’ practices in SSA” (see line 503 in the revised manuscript). 

Minor: 

1. In the Introduction, referencing the increasing number of doctors across Africa per population is important but does not seem specific to the practice of primary care. To strengthen the point, it would be helpful to include the shift towards generalism (vs specialty training) of doctors (if this is true?) as well as overall statistics from SSA, rather than only Benin and the DRC.

Our response

We included overall statistics on the trends in the physicians' ratio in SSA (see lines 74 to 75 in the revised manuscript). The data on generalist physicians in SSA is scattered and poorly available (see, for example, https://apps.who.int/gho/data/node.main.HWFGRP_0020?lang=en). However, ad-hoc observations and some previous research pointed to the increasing presence of medical doctors at the primary care level, especially in the private sector (1,2,6). 

 2. There are many tables, figures, a map, supplementary materials, etc. that are rather overwhelming. I wonder whether some of these might be combined or omitted entirely? 

I would suggest moving Table 1 (the search strategy) into an appendix, and integrating the information from Figure 2 elsewhere. 

Our response

We deleted the map and the table presenting the countries where each type of PCPs has been reported (table 4 in the original manuscript), as the information in these two elements is already presented in the supporting information file S5 table (list of paper included) and in table 2 in the revised manuscript (characterisation of the papers included). We have also moved the search strategy into the supporting information. 

The numbers in Table 3 do not add up (total of 78), so it needs to be stated if there were multiple designations of doctors studied in a single paper.

Our response

Indeed, there were multiple designations of doctors in some papers. We specified this under table 3. 

I would also recommend trying to combine information from Tables 3 and 4 with Figure 2 in some way (so the map could stay with the particular countries appropriately labelled). The fragmentation of information makes the paper seem more complex than it is and difficult to follow.

Our response

We deleted the table 4 of the original manuscript as the information there can be easily retrieved from the supporting information file S5 table which presents the country each document refers to and the name given to the PCPs in the document. We also deleted the map as the information on this map is now in table 2 in the revised manuscript (characterisation of the papers included).

Table 5 columns should be flipped: “Themes” on the left and “Dimensions” on the right. I am also not sure these are the correct labels for the headings (another French to English ‘lost in translation’?) because, as stated above, the data extraction form is not available. 

Our response

We flipped the columns as suggested. As explained now in the methods section from line 161 to 172 and in the results section from line 230 to 231, we grouped some themes into broader themes, based on the strong relationship we found between them. It is these broader themes that we have called "dimensions". 

We provide the data extraction form as a supporting file. 

3. NB: Libya not in SSA. It might be useful to indicate or reference your source(s) for all of the countries designated in your search strategy as being in SSA (eg: UNESCO; World Bank; etc.) 

Our response

We included Libya by mistake. The list was obtained from brainstorming and discussions between the authors. After the reviewer's comment, we carefully reviewed the list, based on the World Bank classification of countries. We removed Libya. Other mistakes were not found. 

4. While discussing the division of labour in the manuscript (who did what), you might want to consider indicating this by including the authors’ initials. 

Our response

We included this in the manuscript (see lines 139, 141, 143, 144 and 145). 

 5. Once the issues above are addressed, the abstract will require revision. Either way, the 'PCC' should be made clear in the introduction; the Methods section would benefit from synthesis and alignment with the text of the manuscript; the Findings should reflect only what is included in the 73 articles under review; similarly, the conclusion can only draw on what is published and/or identified as gaps in the literature. I am a bit worried about some of the quantitative declarations like, "the increasing presence of medical doctors at the first line of health care" or the assertion that GPs "constitute the bulk of FLMDs in SSA" when it is not clear that this information is derived from the scoping review. See comments above about possible over-stating the concluding points. Once again, a scoping review can only draw conclusions about what is already published, or comment on research gaps. 

Our response

We revised the abstract, and we carefully avoided over-stating conclusions (see the abstract section from line 27 to 53 in the revised manuscript). 

6. The term “professional identity” within medicine has been richly explored. Lines 171 – 176 would benefit from citing some of this literature, rather than the generic occupational reference #28. This is important to adequately set up what follows later regarding the “Professional Identity of FLMDs” in lines 196 – 249.

Our response

Thank you for the suggestion. We used 2 more references, specific to the medical field, to better characterize the professional identity and support the rationale for grouping some themes under this dimension (see 232 to 242 and references N° 34 and N° 35 in the revised manuscript). 

 7. There is only brief mention of limitations on data quality. This could be addressed in more detail.

Our response

We expanded this more in the limitations section. We highlighted the fact that, given the lack of appraisal of the data quality and the fact that we did not comprehensively search for grey literature, the conclusions of this review still need to be confirmed by good empirical studies (see line 622 to 625 in the revised manuscript). 

8. References: I am curious about why DOIs were not routinely included. 

Our response

We corrected the references list by including the DOIs. 

9. There are several categorical statements in the manuscript which are clearly false, or end up being contradicted later. One of these being line 237: “The GPs […] are medical doctors with a mere undergraduate training.” Unfortunately, this is simply not true. Most countries require some form of vocational training following an undergraduate degree, prior to working as a general practitioner or medical officer. This could be a rotating one or two year internship as well as an additional year of community service. As an example of conflicting information, lines 277 – 278 “…most of the MGCs work in CHCs, which are public entities…” contradicts line 224 which states “The MGCs work in the private sector and claim not-for profit status.” Table 6 is also prone to such errors (eg: FPs “do not provide first-contact care”), and should be reviewed. A final careful read-through of the text would be important to address such discrepancies. 

Our response

Thank you for the comments. We reviewed the wordings in the manuscript for more clarity and to avoid generalised conclusions. For instance, we clarified the status of the community health centres in Mali. They are not public entities indeed, since the State has devolved their management to local health associations. For more clarity, we rephrased the sentence as follows: "… most MGCs work in community health centres, which are fully integrated into the district health map even if their management has been devolved to local health associations..." (see line 355 to 357 in the revised manuscript). 

We also clarified that, when we refer to the postgraduate training of the PCPs, we refer to formal training that addresses the principles and specificities of primary care and that can prepare the PCPs to work at the first-line (see line 217 to 218 in the revised manuscript). While such training exists for the family physicians and the "médecins généralistes communautaires", the PCPs categorized as “GPs” in this review are either the doctors for whom the papers reviewed indicate that this training does not exist or the doctors for whom the existence of this training is not clear. This again indicates the need for better research on this category. In Benin, for example, apart from an internship in a rural area during the sixth year of the undergraduate studies, there is no other requirement, once the diploma is obtained, for the doctor to start practising. But, this is not necessarily the case in other countries. 

We have carefully checked for other discrepancies throughout the document.

10. While not the focus of this current study, one wonders about the impacts of migration, brain drain, the climate crisis and regional conflicts on the subject under review. Even passing mention of these factors might be in order in the discussion given how prevalent and weighty these are for the SSA context within which this review takes place. 

Our response: 

Thank you for these relevant suggestions. In the revised manuscript, we discussed how inadequate governance of the PCPs might lead to internal or external brain drain (see lines 567 to 572 in the revised manuscript). We have also highlighted the need to pay attention to the role the PCPs could play in addressing the emerging challenges that Africa is currently facing, including COVID-19 and climate changes (see line 653 to 655 in the revised manuscript). 

References

1. Dugas S, Van Dormael M. La construction de la médecine de famille dans les pays en développement. W. Van Lerberghe, G. Kegels, editors. Antwerp: ITGPress; 2003. French. http://dspace.itg.be/bitstream/handle/10390/1526/shsop22.pdf?sequence=1. 

2. Desplats D, Koné Y, Razakarison C. Pour une médecine générale communautaire en première ligne. Med Trop. 2004;64(6):539-44. French. DOI: 10.1055/s-0029-1237558.

3. Hellenberg DA, Gibbs T, Megennis S, Ogunbanjo GA. Family medicine in South Africa: where are we now and where do we want to be? Eur J Gen Pract. 2005;11(3):127-30. DOI: 10.3109/13814780509178253.

4. Moosa S, Mash B, Derese A, Peersman W. The views of key leaders in South Africa on implementation of family medicine: critical role in the district health system. BMC Fam Pract. 2014 Jun;15(125):1-7. DOI: 10.1186/1471-2296-15-125. 

5. Makwero M, Lutala P, McDonald A. Family medicine training and practice in Malawi: history, progress, and the anticipated role of the family physician in the Malawian health system. Malawi Med J. 2017 Dec;29(4):312-6. DOI: 10.4314/mmj.v29i4.6. 

6. Willcox ML, Peersman W, Daou P, Diakité C, Bajunirwe F, Mubangizi V, et al. Human resources for primary health care in sub-Saharan Africa: progress or stagnation? Hum Resour Health. 2015;13(76):1-11. DOI: https://doi.org/10.1186/s12960-015-0073-8.

---

## [Decision Letter · Decision Letter 1]

28 Aug 2021

PONE-D-20-25494R1

The expanding movement of primary care physicians operating at the first line of healthcare delivery systems in sub-Saharan Africa: A scoping review

PLOS ONE

Dear Dr. Bello,

Thank you for submitting your manuscript to PLOS ONE. After careful consideration, we feel that it has merit but does not fully meet PLOS ONE’s publication criteria as it currently stands. Therefore, we invite you to submit a revised version of the manuscript that addresses the points raised during the review process.

We look forward to receiving your revised manuscript.

Kind regards,

Virginia E. M. Zweigenthal

Academic Editor

PLOS ONE

Journal Requirements:

Additional Editor Comments (if provided):

Dear Colleague,

Many thanks for your resubmission. The article is almost ready for publication. We would like you to consider the last comments made by Reviewer 3 and submit changes.

Many thanks,

Reviewers' comments:

Reviewer's Responses to Questions

**Comments to the Author**

1. If the authors have adequately addressed your comments raised in a previous round of review and you feel that this manuscript is now acceptable for publication, you may indicate that here to bypass the “Comments to the Author” section, enter your conflict of interest statement in the “Confidential to Editor” section, and submit your "Accept" recommendation.

Reviewer #1: All comments have been addressed

Reviewer #3: All comments have been addressed

2. Is the manuscript technically sound, and do the data support the conclusions?

Reviewer #1: Yes

Reviewer #3: Yes

3. Has the statistical analysis been performed appropriately and rigorously? 

Reviewer #1: N/A

Reviewer #3: N/A

4. Have the authors made all data underlying the findings in their manuscript fully available?

Reviewer #1: Yes

Reviewer #3: Yes

5. Is the manuscript presented in an intelligible fashion and written in standard English?

Reviewer #1: Yes

Reviewer #3: Yes

6. Review Comments to the Author

Reviewer #1: Thank you for the opportunity to review the revised manuscript. I am satisfied with the changes made.

Reviewer #3: The rigorous and detailed reviews during the first round of the process are appreciated, as is the thorough response by the authors and the extensive reworking of the manuscript. The quality of the manuscript has improved considerably and, in particular, the engagement around the nomenclature of the primary care physicians and the categories that were developed as part of the analysis made a substantial difference.

One of the responses to the reviewer regarding the question of authorship to include authors from Africa was well presented in the response to the reviewer, and it would strengthen the paper if a statement to that effect either is included as 'positionality' in the manuscript or, perhaps more appropriately, as a postscript at the end of the paper. The current discussions regarding decoloniality and authorship of research from the global south in the field of global health is arguing for a more transparent positioning.

7. PLOS authors have the option to publish the peer review history of their article (what does this mean?). If published, this will include your full peer review and any attached files.

Reviewer #1: No

Reviewer #3: No

---

## [Author Response · Author response to Decision Letter 1]

7 Sep 2021

Response to the reviewer 3

Review comment

One of the responses to the reviewer regarding the question of authorship to include authors from Africa was well presented in the response to the reviewer, and it would strengthen the paper if a statement to that effect either is included as 'positionality' in the manuscript or, perhaps more appropriately, as a postscript at the end of the paper. The current discussions regarding decoloniality and authorship of research from the global south in the field of global health is arguing for a more transparent positioning.

Our response 

Thank you very much for reviewing our manuscript and for your valuable comment. The suggestion to include a statement on the authors' positionality was an important contribution to our work. It helped further to position the paper in the current global health debates. 

As suggested, we added a postscript on the authors' positionality (see line 656 to line 705 in the revised manuscript).

Response to the issues raised by the editorial office

Sincere apologies to the issues related to the numbering of the tables. In fact, we deleted a table from the first submission. This led to this error in the numbering. Please find below how we have addressed the issues in the revised manuscript.

Issue 1

Please ensure that you refer to Table 4 and table 5 in your text as, if accepted, production will need this reference to link the reader to the Table. 

Our response

We now referred to table 4 and table 5 on lines 230 and 495 respectively.

Issue 1

Please include a copy of Table 6 which you refer to in your text on page 27.

We meant to refer to table 5, not table 6. We corrected this in the revised manuscript, line 495.

---

## [Decision Letter · Decision Letter 2]

11 Oct 2021

The expanding movement of primary care physicians operating at the first line of healthcare delivery systems in sub-Saharan Africa: A scoping review

PONE-D-20-25494R2

Dear Dr. Bello,

We’re pleased to inform you that your manuscript has been judged scientifically suitable for publication and will be formally accepted for publication once it meets all outstanding technical requirements.

Kind regards,

Virginia E. M. Zweigenthal

Academic Editor

PLOS ONE

Additional Editor Comments (optional):

Dear Authors,

We are delighted to inform you that your article on the expansion of primary care physicians working in SSA has been accepted for publication. We think that the review highlights the various roles of primary care physicians bringing research together in a meaningful way.

Many thanks,

Dr Virginia Zweigenthal

Reviewers' comments:

Reviewer's Responses to Questions

**Comments to the Author**

1. If the authors have adequately addressed your comments raised in a previous round of review and you feel that this manuscript is now acceptable for publication, you may indicate that here to bypass the “Comments to the Author” section, enter your conflict of interest statement in the “Confidential to Editor” section, and submit your "Accept" recommendation.

Reviewer #3: All comments have been addressed

2. Is the manuscript technically sound, and do the data support the conclusions?

Reviewer #3: Yes

3. Has the statistical analysis been performed appropriately and rigorously? 

Reviewer #3: N/A

4. Have the authors made all data underlying the findings in their manuscript fully available?

Reviewer #3: Yes

5. Is the manuscript presented in an intelligible fashion and written in standard English?

Reviewer #3: Yes

6. Review Comments to the Author

Reviewer #3: The queries I had raised in the previous round of review were addressed adequately, particularly to the thorough first round of review.

7. PLOS authors have the option to publish the peer review history of their article (what does this mean?). If published, this will include your full peer review and any attached files.

Reviewer #3: **Yes: **Bernhard Gaede

---

## [Editor Report · Acceptance letter]

13 Oct 2021

PONE-D-20-25494R2 

The expanding movement of primary care physicians operating at the first line of healthcare delivery systems in sub-Saharan Africa: A scoping review 

Dear Dr. Bello:

I'm pleased to inform you that your manuscript has been deemed suitable for publication in PLOS ONE. Congratulations! Your manuscript is now with our production department. 

Kind regards, 

on behalf of

Dr. Virginia E. M. Zweigenthal 

Academic Editor

PLOS ONE